

# A 148-year precipitation oxygen isoscape for China generated based on data fusion and bias correction of iGCMs simulations

Jiacheng Chen [1], Jie Chen [1,2], Xunchang J. Zhang [3], Peiyi Peng [4], Camille Risi [5]

[1]State Key Laboratory of Water Resources & Hydropower Engineering Science, Wuhan University, Wuhan, 430072, China
[2]Hubei Key Laboratory of Water System Science for Sponge City Construction, Wuhan University, Wuhan, 430072, China
[3]USDA-ARS, Grazinglands Research Laboratory, 7207W. Cheyenne St., El Reno, OK 73036, USA
[4]Chongqing Southwest Research Institute for Water Transport Engineering, Chongqing Jiaotong University, Chongqing, 400016, China
[5]Laboratoire de Meteorologie Dynamique, IPSL, CNRS, Ecole Normale Superieure, Sorbonne Universite, PSL Research
University, Paris, France

*Correspondence to*: Jie Chen (jiechen@whu.edu.cn)

**Abstract.** The precipitation oxygen isotopic composition is a useful environmental tracer for climatic and hydrological studies. However, the observed precipitation oxygen is limited at both temporal and spatial scales. Isotope-equipped general circulation models (iGCMs) can compensate for the temporal and spatial discontinuity of observation networks, but they
suffer from coarse spatial resolutions and systematic biases. The objective of this study is to build a high-resolution precipitation oxygen isoscape in China for a period of 148 years by integrating observed and iGCMs-simulated precipitation oxygen isotope composition ($\delta^{18}O_p$) using data fusion and bias correction methods. The temporal and spatial resolutions are month and 50-60 km for the isoscape, respectively. Prior to building the oxygen isoscape, the performance of two bias correction methods (BCMs) and three data fusion methods (DFMs) is compared after post-processing of eight iGCM
simulations. Results show that the outputs of the Convolutional Neural Networks (CNN) fusion method exhibit the strongest correlation with observations with correlation coefficient mostly larger than 0.8, and the smallest bias with root mean square error mostly smaller than 2‰. The other two DFMs also perform slightly better than the two BCMs, which show similar performance. Thus, precipitation oxygen isoscape is generated by using the CNN fusion method for the 1969-2007 period in which all iGCMs have output and by using the bias correction methods for the remaining years. Based on the precipitation
oxygen isoscape, the spatiotemporal patterns of $\delta^{18}O_p$ across China are investigated. The generated isoscape shows similar spatial and temporal distribution characteristics to observations. In general, the distribution pattern of $\delta^{18}O_p$ is consistent with the temperature effect in northern China, and with the precipitation amount effect in southern China, and be more specific in the Qinghai-Tibet Plateau of China. The $\delta^{18}O_p$ time series mirrors a fluctuating upward trend of the temperature or precipitation in most regions of China. The temporal and spatial distribution characteristics of the generated isoscape are
consistent with the characteristics of atmospheric circulation and climate change, indicating successful assimilation and extension of the observed precipitation oxygen isotopes in time and space. Overall, the built isoscape is reliable and useful for providing strong support for tracking atmospheric and hydrological processes. The dataset is available in Zenodo at https://doi.org/10.5281/zenodo.5703811 (Chen et al., 2021).



## 1 Introduction

Stable oxygen and hydrogen isotopes are the components of water molecules in natural water bodies. The fractionation of stable isotopes, namely, the distribution of stable isotopes in different ratios between two phases, occurs during each phase transition in the water cycle (Dansgaard, 1964). Therefore, stable isotopes are very sensitive to environmental changes and can record the internal process of water cycles, providing an effective tracking method in the study of complex hydrological and climatic processes (Gibson et al., 2005; Gat, 1996; Galewsky et al., 2016; Ansari et al., 2020).

40        The stable isotopes in precipitation are generally obtained from station measurements. In 1961, the International Atomic Energy Agency (IAEA) and the World Meteorological Organization (WMO) launched the Global Network of Isotopes in Precipitation (GNIP). Due to the scarcity of stations on the Tibetan Plateau, the Chinese Academy of Sciences (CAS) launched the Tibetan Plateau Network of Isotopes in Precipitation (TNIP) in 1991 (Yu et al., 2016a). However, most Chinese stations in GNIP stopped monitoring in the early 2000s (Zhang and Wang, 2016). In order to continue the systematic study,

the CAS established the Chinese Network of Isotopes in Precipitation (CHNIP) based on the Chinese Ecosystem Research Network (CERN) in 2004. Due to the difficulty and high cost of measuring precipitation isotope ratios (Allen et al., 2018), most of the observed data are short in length. The spatial distribution of observation stations is uneven, with few stations in inaccessible areas (Wang et al., 2015).

        The stable isotopes in precipitation can also be simulated by isotope-equipped general circulation models (iGCMs). In

contrast to observations, iGCMs can provide time-continuous and space-regular isotope data. The general circulation model (GCM) mainly describes the temporal and spatial changes of physical variables through physical equations, and the essence of iGCMs is to introduce the cycle of water stable isotopes into each stage of the water cycle in GCMs (Xi, 2014). Driven by the external climate boundary conditions, iGCMs can provide the composition of precipitation stable isotopes by calculating the basic mass, momentum, energy and water balance equations of air columns and reasonably parameterizing the basic

meteorological processes (Sturm et al., 2010). Because the initial conditions, boundary conditions and process parameters of each model are different, the outputs of each iGCM are various. The Stable Water Isotope Intercomparison Group (SWING) and the second phase of SWING have been established to have a comprehensive comparison among different iGCMs and the related isotope measurements (Zhang et al., 2012). On this basis, the comparison and evaluation of iGCMs have been conducted in many studies (Zhang et al., 2012; Wang et al., 2015; Risi et al., 2012; Che et al., 2016). Even though iGCMs

have reliable physical bases and can reflect the dynamic characteristics of water stable isotopes, they are computationally expensive and require thorough validation (Sturm et al., 2010; Galewsky et al., 2016). Moreover, the spatial resolution of iGCMs is on the order of hundreds kilometres, which is too coarse to investigate the water cycle for a specific region or watershed. Furthermore, outputs of iGCMs are usually subjected to systematic bias when compared with gauged observations, and there is no single iGCM that consistently performs better than the others (Zhang et al., 2012; Wang et al.,

2015; Conroy et al., 2013). The use of ensemble mean of multiple iGCMs is a desirable approach for better use of information available from all iGCMs.

Overall, both observations and iGCM simulations have advantages and disadvantages. The effort of constructing a database by taking advantages and circumventing disadvantages of both becomes a challenge. Accordingly, in order to make full use of the advantages of high quality observations and continuous iGCM simulations, this study aims at constructing a
high-resolution precipitation oxygen isoscape in mainland China by data fusion and bias correction of iGCM simulations. In order to determine the best scheme to build the dataset, two bias correction methods (BCMs) and three neural network data fusion methods (DFMs) are first compared in terms of bias correcting and fusing iGCM simulations. The new isoscape in monthly temporal and approximately 0.5° spatial resolutions is produced by combining the Convolutional Neural Networks (CNN) fusion method and bias correction method for the 1870-2017 period. The spatial and temporal distribution
characteristics of oxygen isotopes in precipitation are then analysed for China.

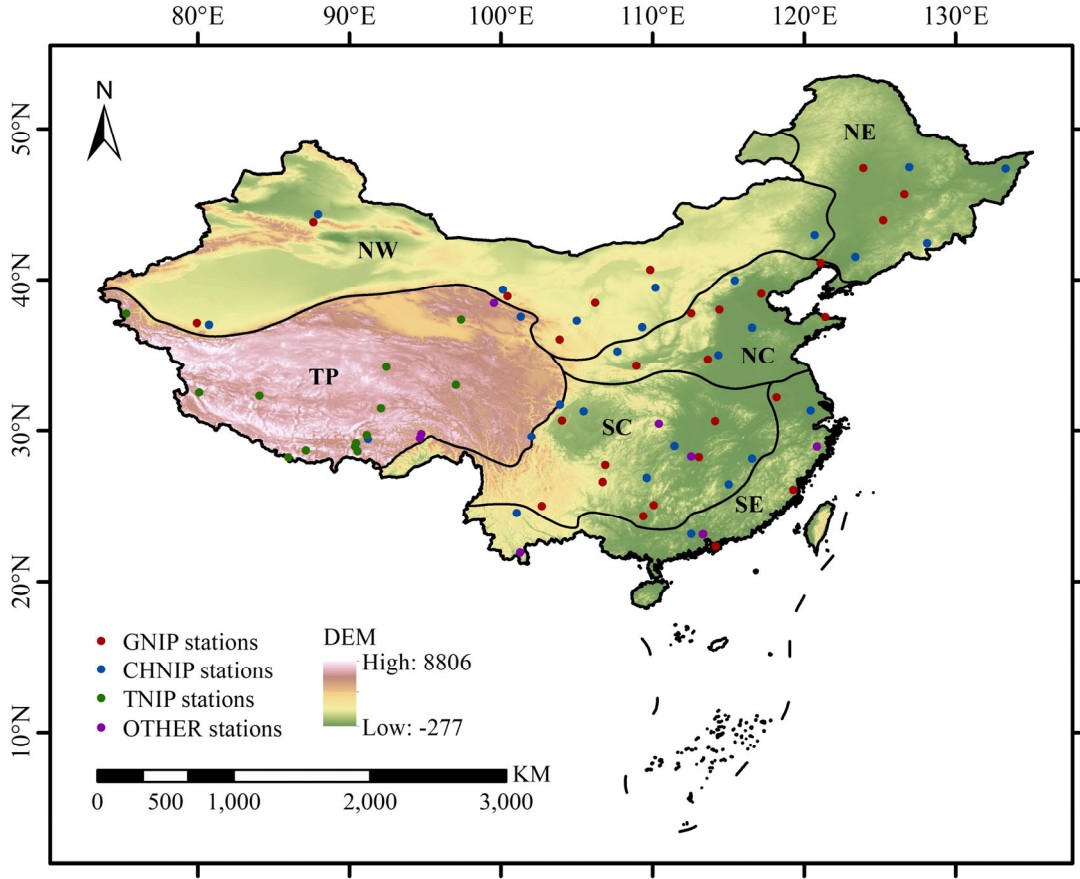

**Figure 1. Map of the station locations and topography in mainland China. The dots indicate the distribution of isotope observation stations, with different colours representing different sources. The six sub-regions are plotted (NE – Northeast China, NC – North China, SC – South China, SE – Southeast China, TP – Tibetan Plateau, NW – Northwest China).**



## 2 Study area and data

### 2.1 Study area

China is located in the east of Eurasia and on the west coast of the Pacific Ocean. The topography of China generally presents three steps descending to the east. The climate is complex and diverse in China. Heavily influenced by the continents and oceans, the monsoon climate is significant, especially for the east of China. The spatiotemporal variation of
precipitation stable isotopes are very complex due to the significant changes in winter and summer circulation (Liu et al., 2014). Mainland China can be geographically classified into three sub-regions, the eastern monsoon region, the arid northwest region (NW) and the Qinghai-Tibet Plateau region (TP), according to topography, climate, soil and vegetation. According to temperature and precipitation, the eastern monsoon region is further divided into two sub-regions: north (NE and NC) and south (SC and SE). In addition, the southern part of the eastern monsoon region is further divided into southeast
coastal region and inland region, considering the characteristics of observed isotope data. To sum up, the study area is divided into six sub-regions for our analysis: Northeast China (NE), North China (NC), South China (SC), Southeast China (SE), Qinghai-Tibet Plateau (TP) and Northwest China (NW), as shown in Fig. 1.

### 2.2 Datasets

There are 78 oxygen isotope observation stations in the study area (Fig. 1), including 29 GNIP stations (available at
https://nucleus.iaea.org/wiser), 27 CHNIP stations (Liu et al., 2014), 13 TNIP stations (Yao et al., 2013) and 9 stations from other sources (mainly from references). Monthly oxygen isotope composition of precipitation ($\delta^{18}O_p$) is used for analysis. The time span of GNIP data mostly ranges from 1980 to 2000, and the length of the time period is basically 5-15 years. For CHNIP, most time periods are about 2-5 years ranging from 2005 to 2010. Most TNIP data were between 1995 and 2005, with varying lengths.
Eight $\delta^{18}O_p$ spatio-temporal fields simulated by five iGCMs (CAM2, GISS E, HadAM3, LMDZ4 and MIROC32) are used, which are selected from the SWING2 archive (available at https://data.giss.nasa.gov/swing2/). Six of eight simulations are free-running, forced only by observed sea surface conditions. The remaining two (GISS E and LMDZ4) were nudged to constrains of large-scale atmosphere circulation, so that the dynamical fields in simulations are close to the observations (Yoshimura et al., 2008). In addition to SWING2 simulations, a zoomed simulation by LMDZ4, with the horizontal
resolution of 50-60km (Risi et al., 2010; Gao et al., 2011) and nudged by reanalyses, is also used. Detailed information about these iGCMs can be found in Table 1.

**Table 1. Time periods and basic outputs information of selected iGCMs.**

| GCM | Simulation method | Horizontal resolution (longitude × latitude) | Time period | Key references |
|---|---|---|---|---|
| CAM2 | Free-running | 2.8° × 2.8° | 1958-2003 | Lee et al. (2007) |
| GISS E | Free-running and | 2.5° × 2° | 1969-2009 | Schmidt et al. (2007) |





| GCM | Simulation method | Horizontal resolution (longitude × latitude) | Time period | Key references |
|---|---|---|---|---|
| HadAM3 | nudged by NCEP Free-running | 3.75° × 2.5° | 1870-2001 | Tindall et al. (2009) |
| LMDZ4 | Free-running and nudged by ECMWF | 3.75°×2.5° | 1979-2007 | Risi et al. (2010) |
|  | Zoomed (nudged by ECMWF) | 50-60 km | 1979-2017 | Gao et al. (2011) |
| MIROC32 | Free-running | 2.8°×2.8° | 1979-2007 | Kurita et al. (2011) |

The time span of the isoscape built in this study covers the union set of all simulations ranging from 1870 to 2017. Since the temporal lengths of eight iGCM are not identical, the number of iGCM simulations used to build the isoscape varies. Specifically, for 1979-2001, a total of eight simulations from all five iGCMs were used; for 2002-2007, six simulations from three iGCMs (GISS E, LMDZ4 and MIROC32) were used; and for 1969-1978, four simulations from three iGCMs (CAM2, GISS E and HadAM3) were used. For the remaining periods, there is only one simulation or two: for 1958-1968, CAM2 and HadAM3 were used; for 1870-1957, HadAM3 was used; and for 2008-2017, LMDZ4 zoomed was used.

The stable isotope composition of precipitation is expressed in the relative permillage (‰) derived from the standard sample (Clark and Fritz, 1997) as:

$$\delta = \left( \frac{R_{\text{sample}}}{R_{\text{V-SMOW}}} - 1 \right) \times 1000‰ \tag{1}$$

where $R$ is the ratio of heavier isotope to common isotope ($^{18}O/^{16}O$), and the subscripts sample and V-SMOW represent standard sample and Vienna Standard Mean Ocean Water, respectively.

## 3 Methods

### 3.1 Bias correction methods

BCMs aim to correct the mean, variance and/or quantile of the climate model time series, so that the corrected model time series can better match those of the observations (Maraun, 2013). In this study, two typical methods (i.e. linear scaling (LS) and distribution translation (DT)) were used to correct the bias of iGCM at the monthly timescale. These two BCMs can be classified into mean-based scaling (i.e. LS) and distribution-based correction (i.e. DT) approaches (Chen et al., 2013). The mean-based scaling uses a constant correction factor for the entire time series, while the distribution-based approach uses correction factors that vary with the quantiles of the distribution (Chen et al., 2016).

The LS method is the simplest bias correction method. The differences between observations and raw iGCM simulations were applied to simulations to obtain the bias-corrected isotope time series for each season and sub-region. Specifically, for a particular sub-region, the differences (defined as correction factors) in mean values between observed and simulated isotopes



were first calculated at the seasonal basis using Eq. (2). The calculated correction factors were then applied to simulated isotopes for the entire period using Eq. (3).

$$R_{\mathrm{LS}} = \overline{\delta O}_{\mathrm{obs,s,sr}}^{\mathrm{ref}} - \overline{\delta O}_{\mathrm{raw,s,sr}}^{\mathrm{ref}} \tag{2}$$

$$\delta O_{\mathrm{cor,s,sr}} = \delta O_{\mathrm{raw,s,sr}} + R_{\mathrm{LS}} \tag{3}$$

where $R_{\mathrm{LS}}$ is the correction factor; the superscript ref represents the reference period; the subscripts obs, raw and cor represent observations, raw simulations and corrected simulations; and s and sr represent a specific season and a sub-region.

The implementation of the DT method is similar to the LS method. However, the differences (i.e. correction factors) between observed and simulated isotopes were calculated for each of 100 integral percentiles as shown in Eq. (4) – (6), to represent the distribution for each season in each sub-region. The correction factors of grid points were obtained by

interpolating or extrapolating the factors of observation stations using Eq. (5).

$$R_{\mathrm{DT}}^{\mathrm{ref}} = \delta O_{\mathrm{obs,q}}^{\mathrm{ref}} - \delta O_{\mathrm{raw,q}}^{\mathrm{ref}} \tag{4}$$

$$R_{\mathrm{DT}}^{\mathrm{ref}} \xrightarrow{\text{Interpolation/Extrapolation}} R_{\mathrm{DT}} \tag{5}$$

$$\delta O_{\mathrm{cor,s,sr}} = \delta O_{\mathrm{raw,s,sr}} + R_{\mathrm{DT}} \tag{6}$$

where the subscript q is a percentile for a specific season in a sub-region. Other superscripts and subscripts are the same as

Eq. (2) and (3).

### 3.2 Neural network data fusion

The neural network is a kind of mathematical model, which imitate the behaviour characteristics of animal neural network and carry out distributed parallel computing (Rumelhart et al., 1994; Krenker et al., 2011). Performing calculations and spreading information through large numbers of interconnected neurons, neural networks are often used to describe complex

relationships between inputs and outputs, or to explore the internal structure and patterns of data (Hsu et al., 1995; French et al., 1992). In this study, Back Propagation Neural Network (BP), Long Short-Term Memory (LSTM) Neural Network and Convolutional Neural Network (CNN) are adopted for data fusion, considering BP's simplicity and practicality, LSTM's advantages in time series prediction and CNN's outstanding performance in various fields. The structures of these three neural network DFMs are presented in Fig. 2.

BP, first proposed by Rumelhart et al. (1986), is a multilayer feed-forward network trained by the error back-propagation algorithm. BP is one of the most widely used neural network models with high simulation accuracy for nonlinear functions. The main characteristic of BP is that the input signal is processed layer by layer from the input layer to the hidden layer and then to the output layer, and each neuron carries out the weighted sum of the input signal through the activation function. If the error between the actual output and the expected output is larger than the set value, the weight and bias of the network

are continuously corrected by the backpropagation to minimize the loss function. The isotope simulations of iGCMs were selected as the input of BP and the observations as the expected output to calculate the loss function. The input layer was the

corresponding input parameter, and the output layer was the fusion isotope value. The hidden layer had two layers with 16 and 4 neurons respectively. The activation function was the Rectified Linear Unit (ReLU). The Stochastic Gradient Descent (SGD) was used to optimize the loss function iteratively. Mean Square Error (MSE) was chosen as the loss function. The maximum number of iterations was 1000, and the training stopped if the loss function did not decrease within 150 epochs.

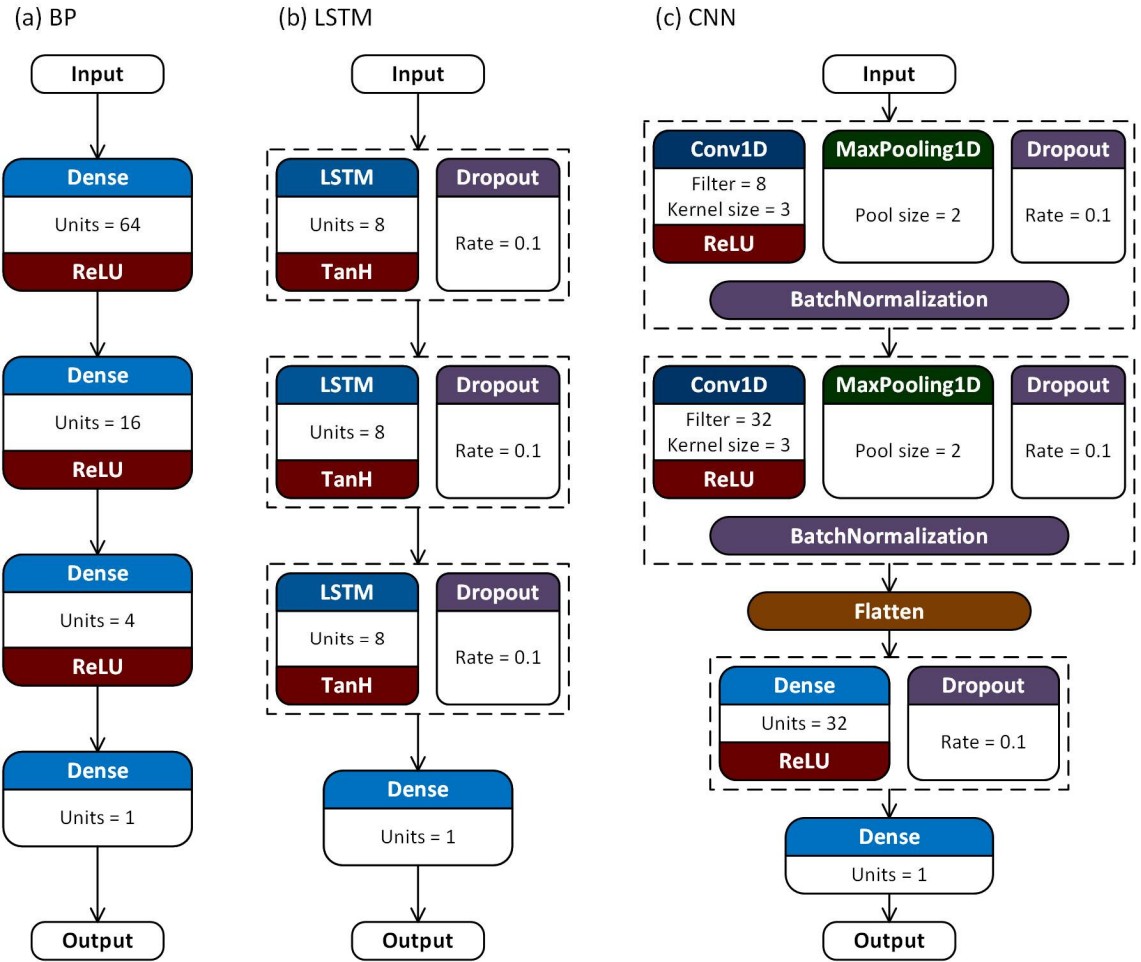

**Figure 2. Network structure of BP, LSTM and CNN fusion method.**

LSTM is very efficient for sequential data and is derived from Recurrent Neural Network (RNN) with memory function. RNN has a sequential feed-forward connection, so that the information of the past moment can affect the output of the present moment (Zhang et al., 2018). The traditional RNN has the problems of vanishing gradient and exploding gradient (Hochreiter, 1998). To solve these problems, the LSTM Neural Network was proposed (Hochreiter and Schmidhuber, 1997). A basic LSTM neuron usually consists of a memory cell and three gates (i.e. input gate, forget gate and output gate). Memory cells are used to store past information, realizing long-distance dependent learning of sequence features. The input gate determines which inputs are saved to the cell; forget gate determines what information is retained from the previous





moment; output gate determines what information needs to be output. In this study, a fully connected layer was added after
three LSTM layers to generate fusion results. The number of neurons in the LSTM layer was set to 8. Dropout layers were
applied to the three LSTM layers of the network with the probability of 0.1 to make the model more robust. The activation
function used the most common setting in LSTM, the Hyperbolic Tangent (TanH) function. The optimizer of the model was
set as the Adaptive Moment Estimation (Adam) algorithm, which performs optimally. Similar to BP, MSE was chosen as the

loss function. The maximum number of iterations was 1000 with the patience of 100 epochs.

    CNN was first proposed by Lecun (1989), for the problem of handwritten digit recognition. CNN combines three
advantages of local connectivity, weight sharing and pooling. On one hand, it reduces the number of weights, making the
network easy to optimize. On the other hand, it reduces the complexity of the model and alleviates the overfitting problem. It
is one of the most widely used neural networks with the best performance. The convolutional layer and pooling layer of the

hidden layer are the core modules of CNN. The function of the convolutional layer is to extract features of the input data by
convolutional kernels. The pooling layer performs feature selection and information filtering on the feature map output by
the convolution layer. In this study, CNN was mainly composed of two convolutional layers and pooling layers. The two
convolutional layers had 8 and 32 convolutional kernels, respectively, and the size of the convolution kernels was 3. The
pooling layers used the max pooling method, and the pooling size was 2. Two fully connected layers were added at the end

with 32 and 1 neurons, respectively, to remove the spatial topology and output the results. The convolutional layer and the
fully connected layer were connected by flattening the output of the convolutional layer through the flatten layer. Batch
normalization layers were inserted into the model to improve the speed, performance and stability of the neural network.
Dropout layers were also applied with the probability of 0.1. The activation function was the ReLU. The optimizer of the
model was the Adam, with the loss function of MSE. The maximum number of iterations was 1000 with the patience of 100

epochs.

### 3.3 Cross-validation experiments

In order to make full use of the data and reduce the variation of model accuracy caused by the difference between the
training set and the test set, K-fold cross-validation was adopted. In the K-fold (K=5 in this study) experiment, the data set
was randomly divided into K groups, and one of them was used as the test set each time, leaving K-1 groups as the training

set. To fully consider the variations of random division, K-fold cross-validation was repeated 100 times.

### 3.4 Generation of isoscape

The performance of BCMs and DFMs were evaluated by using correlation coefficient (CC) and root mean square error
(RMSE) as metrics for the common period of 1969-2007. On the basis of the performance comparison, the optimal method
was selected for each period to generate the oxygen isoscape. For the common period (1969-2007), the isoscape was

generated using the best-performing methods among BCMs or DFMs; while for the rest periods, only the BCM was used
since there is only one or two simulations. When using the BCM, the ensemble mean of all iGCM simulations and two




BCMs were used. To sum up, the generated isoscape presents monthly $\delta^{18}O_p$ on a 50-60km spatial resolution over mainland China for the 1870-2017 period.

## 4 Results

### 4.1 Evaluation of bias correction and data fusion methods

Prior to applying BCMs and DFMs to build the isoscape, the performance of iGCM simulations was evaluated by comparing gauged observations for the 1969-2007 period. Fig. 3 shows the cumulative distribution functions (CDFs) of $\delta^{18}O_p$ for observations and iGCMs simulations in each sub-region. Generally, the CDFs of observed $\delta^{18}O_p$ can be well represented by iGCM simulations for each sub-region, as the observed CDFs distribute in the centre of simulated ones. For specific regions, the envelope of CDFs is the narrowest for NE and SE, indicating that iGCMs perform consistently better for these two regions. For NC, SC and NW, the CDFs of LMDZ4 (free and nudged) simulated $\delta^{18}O_p$ are relatively close to the observations, while other iGCM simulations generally overestimate the $\delta^{18}O_p$. For TP, the differences between CDFs of observed and simulated $\delta^{18}O_p$ are the largest, indicating the large variability of $\delta^{18}O_p$ simulations. This is expected, as climate models generally perform worse for TP than other regions (Zhu and Yang, 2020; Su et al., 2013).

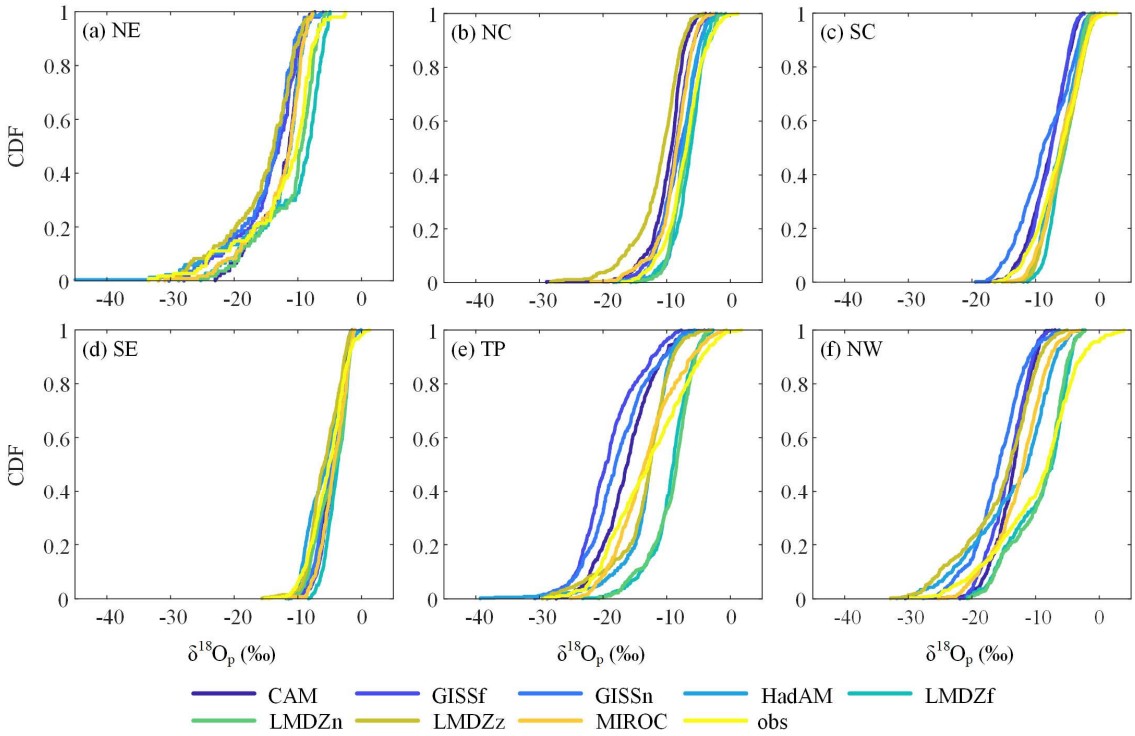

**Figure 3. Cumulative distribution functions of $\delta^{18}O_p$ for eight iGCM simulations in six sub-regions.**

Root mean square error (RMSE) and correlation coefficient (CC) were also calculated to evaluate the accuracy of iGCM simulated $\delta^{18}O_p$. Fig. 4 presents the RMSE and CC for raw iGCM simulations, bias corrected and fused simulations for six sub-regions over the validation periods (1979-2001). Generally, DFMs perform better than BCMs, and both perform better

than raw iGCM simulations. In addition, all the simulations are correlated with the observations with CC ranging between 0.12 and 0.95.

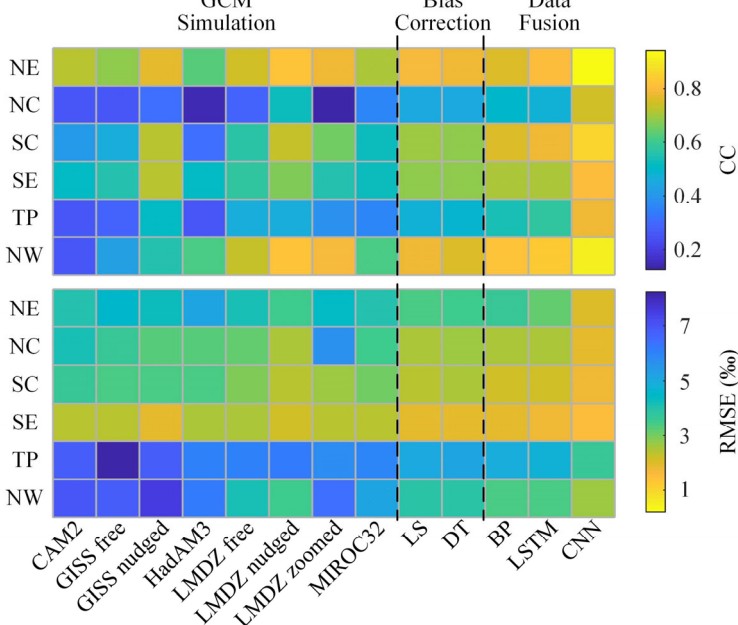

**Figure 4. Average correlation coefficient (CC) and root mean square error (RMSE) metrics of raw, bias-corrected and fused $\delta^{18}O_p$ in six sub-regions.**

The CC and RMSE vary considerably for raw iGCM simulations, with CC ranging from 0.12 to 0.83 and RMSE ranging from 2.0‰ to 9.1‰. The simulations of nudged GISS E and LMDZ4 have the strongest correlation with the observations, and their CC is basically above 0.5, ranging from 0.30-0.83. The error of nudged LMDZ4 is the smallest, and its RMSE ranges between 2.2‰ and 6.8‰.

Generally, the performance of LS and DT is similar, even though the LS method performs slightly better than the DT

method. The CCs range from 0.40 to 0.88 for the LS and DT corrected simulations, with an average increase from 0.52 to 0.64 relative to raw simulations; the RMSEs are between 2.0‰ and 5.7‰, with an average decrease of 23.9%.

For DFMs, BP and LSTM show similar performance, while CNN performs the best. The CC of CNN-generated simulations is all greater than 0.74, increasing from 0.52 to 0.84 on average compared with the raw simulations. The RMSE of CNN-generated simulations is all smaller than 4.0‰, showing 48% reduction relative to the raw simulation on average.





Fig. 4 shows that CC and RMSE of simulations in different sub-regions are quite different. For all raw, fused, and bias corrected simulations, CC is smaller for NC and TP while RMSE is larger for TP and NW than other sub-regions. For DFMs, especially CNN, the differences of CC and RMSE between different regions are smaller.

Generally, all simulations perform worse in NC, TP and NW than in other sub-regions. The poor simulation performance in NC may be due to complex air mass movements (Yang et al., 2016; Peng et al., 2020a), which are difficult to be

accurately simulated by iGCMs. The poor simulation performance in TP and NW may be due to the fact that GCMs cannot accurately describe the atmospheric physical process and simulate precipitation and other meteorological factors in these regions (Miao et al., 2012; Jiang et al., 2016; Chen and Frauenfeld, 2014).

The performance of BCMs and DFMs is also evaluated for three periods (1969-1978, 1979-2001 and 2002-2007) and six sub-regions at a seasonal basis. The seasonal average CC and RMSE for two BCMs and three neural network DFMs are

presented in Figs. 5-6. Generally, all BCMs and DFMs perform very similar for all three periods.

As for CC, the simulations in the northern region (NE, NC, and NW) show strong correlation with observations in spring, while those in the southern region (SC, SE) show strong correlation in summer and autumn. The correlation of CNN fusion simulations is significantly higher than that of the other methods, with CC being mostly above 0.8. The correlation of LS and DT corrected simulations is similar to that of BP and LSTM fusion simulations, with CC being mostly between 0.3 and 0.6,

varying with sub-regions and seasons. The BP and LSTM fusion methods perform slightly worse in NE, but better in TP and NW, compared with BCMs.

As for RMSE, the northern region has a small error of about 2‰ in summer, except for NW with a larger error of 3‰; while the southern region shows little seasonal difference in error, only RMSE in SE being slightly larger in autumn and winter, mostly ranging between 1‰ and 3‰. On the whole, the DFMs perform better than the two BCMs. The errors of

CNN simulations are the smallest in all regions and seasons, which are smaller than 4‰ in TP and NW and mostly smaller than 2‰ in other sub-regions. The errors of BP and LSTM simulations are slightly smaller than those of LS and DT simulations.

From the error bars, the LS and DT methods show smaller uncertainties in CC and RMSE than the BP, LSTM and CNN fusion methods. In addition to the uncertainties brought by cross-validation, the simulations of DFMs also show uncertainties

brought by neural network itself. Generally, CNN fusion simulations show smaller uncertainties than the other two fusion methods.

The above results show that the CNN fusion method consistently performs better than the other methods. To further confirm this conclusion, scatter plots of fused and corrected against observed $\delta^{18}O_p$ are presented in Fig. 7 for the 1979-2001 period. The overall mean CC and RMSE corresponding to the figure are shown in Table 2. It can be seen that the CNN

fusion method shows a stronger correlation with the observations than the other fusion methods and BCMs. The CNN fused $\delta^{18}O_p$ consistently shows the largest CC and smallest RMSE, showing a strong positive linear correlation with the observations with CC being almost all larger than 0.85.





**Figure 5.** Seasonal average results of correlation coefficient (CC) metrics for BCMs and DFMs in six sub-regions. The whiskers denote +/- one standard deviation.





**Figure 6. Seasonal average results of root mean square error (RMSE) metrics (‰) for BCMs and DFMs in six sub-regions. The whiskers denote +/- one standard deviation.**



**Figure 7. Scatter plots of seasonal δ¹⁸Oₚ from bias-corrected and fused output against observations in six sub-regions.**





**Table 2. Correlation coefficient (CC) and root mean square error (RMSE, ‰) metrics corresponding to the Fig. 7.**

| Sub-region | Season | LS | | DT | | BP | | LSTM | | CNN | |
|---|---|---|---|---|---|---|---|---|---|---|---|
| | | CC | RMSE | CC | RMSE | CC | RMSE | CC | RMSE | CC | RMSE |
| NE | SPR | 0.788 | 4.097 | 0.774 | 3.837 | 0.675 | 4.315 | 0.802 | 2.777 | 0.934 | 2.349 |
| | SUM | 0.362 | 1.508 | 0.324 | 1.537 | 0.310 | 1.552 | 0.329 | 1.498 | 0.915 | 0.730 |
| | AUT | 0.559 | 4.099 | 0.513 | 4.258 | 0.382 | 4.562 | 0.333 | 2.889 | 0.913 | 2.091 |
| | WIN | -0.018 | 6.713 | -0.018 | 6.985 | 0.145 | 6.073 | 0.373 | 4.658 | 0.936 | 2.203 |
| NC | SPR | 0.460 | 2.766 | 0.479 | 2.723 | 0.447 | 2.775 | 0.414 | 2.943 | 0.869 | 1.631 |
| | SUM | 0.111 | 2.420 | 0.123 | 2.481 | 0.300 | 2.204 | 0.284 | 2.150 | 0.843 | 1.396 |
| | AUT | -0.029 | 2.797 | -0.011 | 2.795 | 0.200 | 2.457 | 0.268 | 2.527 | 0.849 | 1.422 |
| | WIN | 0.311 | 3.582 | 0.368 | 3.477 | 0.240 | 3.728 | 0.086 | 3.406 | 0.908 | 1.679 |
| SC | SPR | 0.258 | 2.424 | 0.297 | 2.382 | 0.441 | 2.061 | 0.430 | 2.047 | 0.808 | 1.369 |
| | SUM | 0.478 | 2.556 | 0.474 | 2.571 | 0.457 | 2.583 | 0.467 | 2.568 | 0.895 | 1.409 |
| | AUT | 0.499 | 2.557 | 0.539 | 2.516 | 0.599 | 2.351 | 0.613 | 2.350 | 0.895 | 1.340 |
| | WIN | 0.326 | 2.192 | 0.330 | 2.215 | 0.304 | 2.191 | 0.250 | 2.292 | 0.795 | 1.457 |
| SE | SPR | 0.529 | 1.607 | 0.533 | 1.620 | 0.540 | 1.574 | 0.513 | 1.522 | 0.873 | 0.972 |
| | SUM | 0.490 | 1.663 | 0.463 | 1.685 | 0.526 | 1.618 | 0.376 | 1.677 | 0.869 | 1.030 |
| | AUT | 0.434 | 2.806 | 0.438 | 2.926 | 0.488 | 2.728 | 0.491 | 2.810 | 0.869 | 1.624 |
| | WIN | 0.654 | 1.956 | 0.649 | 1.964 | 0.606 | 2.039 | 0.339 | 2.114 | 0.824 | 1.460 |
| TP | SPR | 0.549 | 4.781 | 0.556 | 4.986 | 0.537 | 4.814 | 0.481 | 5.299 | 0.916 | 2.409 |
| | SUM | 0.145 | 5.951 | 0.252 | 6.099 | 0.434 | 5.341 | 0.551 | 5.205 | 0.900 | 2.691 |
| | AUT | 0.121 | 5.731 | 0.164 | 5.921 | 0.355 | 5.209 | 0.315 | 5.380 | 0.864 | 2.852 |
| | WIN | 0.649 | 4.992 | 0.659 | 5.024 | 0.616 | 5.180 | 0.526 | 5.294 | 0.877 | 3.305 |
| NW | SPR | 0.706 | 3.975 | 0.661 | 3.932 | 0.674 | 3.872 | 0.687 | 3.776 | 0.920 | 2.207 |
| | SUM | -0.060 | 3.424 | 0.171 | 3.294 | 0.555 | 2.718 | 0.561 | 2.671 | 0.896 | 1.582 |
| | AUT | 0.536 | 3.746 | 0.481 | 3.908 | 0.643 | 3.440 | 0.574 | 3.553 | 0.929 | 1.735 |
| | WIN | 0.307 | 5.162 | 0.291 | 5.337 | 0.377 | 5.070 | 0.429 | 4.979 | 0.913 | 2.460 |

### 4.2 Spatial variability of precipitation oxygen isotope

Since the above section shows that the CNN method consistently performs better than other methods, it is used to generate precipitation oxygen isotopes for the 1969-2007 period with a spatial resolution of 50-60km on latitude and longitude. Fig. 8

presents the spatial variability of mean $\delta^{18}O_p$ for observed and fused data for the 1969-2007 period. Generally, the CNN fused $\delta^{18}O_p$ shows similar spatial distribution to observations for all four seasons.

In NE, $\delta^{18}O_p$ decreases with increasing latitude for all seasons, and its spatial variation is basically parallel to the latitude, which reflects the latitude effect (Rozanski et al., 1993). Indeed, most of the water vapour in the atmosphere is formed at low latitudes, and Rayleigh distillation continuously depletes the residual water vapour as air masses move toward higher

latitudes, thus depleting the $\delta^{18}O_p$ of the residual water vapour and thus of the rain forming in clouds.



**Figure 8. Seasonal averaged observations (a, c, e, and g) and CNN fused simulations (b, d, f, and h) of δ¹⁸Oₚ in the mainland China.**

In SC and SE, $\delta^{18}O_p$ decreases from the southeast coast to inland. This phenomenon is consistent with the continental effect. As water vapour transfers from the ocean to the interior of the continent, precipitation is formed along the way. The



separation process of heavy isotopes takes place preferentially than that of light isotopes, which leads to the gradual dilution of heavy isotopes in the cloud, and thus makes the proportion of heavy isotopes in the subsequent precipitation lower.

    The $\delta^{18}O_p$ in TP is low in general except for the southeast corner, which is mainly due to the special effect of large landforms (Zhao et al., 2012). The low $\delta^{18}O_p$ in TP is mainly due to its high altitude, with an average altitude being above 4,000m. The moisture in the air mass is gradually removed during the orographic uplift, with heavy isotopes preferred to be

removed during the condensation process, which leads to the dilution of heavy isotopes in water vapour (Rozanski et al., 1993). Higher $\delta^{18}O_p$ in the southeast corner of TP indicates closer vapour sources such as the Bay of Bengal and the Arabian Sea in the Indian Ocean. Due to the terrain barrier of the Himalayas, most of the water vapour can only pass through its southeast corner, along the valley of the major rivers (Nujiang River, Jinsha River, etc.) into the plateau, or through the Yarlung Zangbo River valley into the plateau (Araguás-Araguás et al., 1998).

The $\delta^{18}O_p$ in NW is lower than that in the southern region, but higher than that in NE and TP. This is because NW is far away from the ocean and has a dry climate, so the amount of heavy isotope in water vapour from the ocean is limited. However, a large part of water vapour to generate precipitation in NW comes from terrestrial evaporation (Li et al., 2016). The $\delta^{18}O_p$ in surface water in the arid area is high, resulting in high $\delta^{18}O_p$ in evaporation water vapour and heavy isotope enrichment in precipitation. Another process is re-evaporation of raindrop in arid climate, enriching heavy isotope in

precipitation water. Under the joint control of both, the $\delta^{18}O_p$ in this sub-region varies greatly. In the southern part of the Taklimakan Desert, Xinjiang, $\delta^{18}O_p$ is obviously higher. This is because, the Taklimakan Desert is located in the heart of Eurasia, and it is surrounded by high mountains and has extremely low rainfall. In the southern part of the desert, there is more precipitation in the Kashi-Hotan line, and the water vapour mainly comes from the evaporation of local lakes and rivers, so the ratio of isotopes in precipitation is high.

The seasonality of $\delta^{18}O_p$ varies in sub-regions and is influenced by various factors. For NE and NC, $\delta^{18}O_p$ is lower in winter than in other seasons. NE and NC are influenced by westerly wind and polar continental air mass constantly, with no convergence or strong convection with isotope-depleted air mass. Compared with Pacific air mass, westerly wind and polar air mass are drier and have higher $\delta^{18}O_p$ (Tang et al., 2017). The seasonal distribution pattern of $\delta^{18}O_p$ in NE and NC is consistent with the temperature effect (Dansgaard, 1964). Although the amount effect is not significant at the annual scale in

these regions, it cannot be ignored in the wet season (Yamanaka et al., 2007). In particular, the maximum value of $\delta^{18}O_p$ is observed in spring for NC, when a large part of the precipitation vapour comes from local re-evaporation. The temperature effect is also reflected in NW. Due to the long-term influence of continental air mass, the temperature difference between winter and summer is large, and the $\delta^{18}O_p$ changes synchronously with temperature in these two seasons.

    For SC and SE, $\delta^{18}O_p$ is lower in summer than in other seasons. The climate features of SC and SE are related to the deep

convection driven by the East Asian monsoon (Vuille et al., 2005), which brings water vapour from the Pacific Ocean to eastern China and dominates the sub-regions in summer. Air masses from the Pacific Ocean are more isotopically depleted than those from SC and SE (Zhao et al., 2012; Peng et al., 2020a), so convergence with the Pacific depleted air masses will dilute the isotopic content of precipitation in SC and SE. Therefore, although temperature in summer is generally higher,





depletion of $\delta^{18}O_p$ is usually larger during monsoon season than winter season. The effect of surface temperature on isotopic

fractionation during precipitation is masked by the effect of precipitation amount (Araguás-Araguás et al., 1998). The temporal distribution pattern of $\delta^{18}O_p$ in SC and SE is influenced by heavy monsoon precipitation and follows the amount effect (Rozanski et al., 1993; Dansgaard, 1964).

In TP, $\delta^{18}O_p$ is positively correlated with temperature in the non-monsoon region (northern part of the plateau), with high $\delta^{18}O_p$ in summer and low in winter, reflecting the temperature effect. For the monsoon region (southern part of the plateau),

$\delta^{18}O_p$ is high in winter and spring and low in summer and autumn, which is obviously influenced by marine air mass and shows obvious amount effect. These results are similar to previous studies (Yu et al., 2021; Yao et al., 2013).

### 4.3 Temporal variability of precipitation oxygen isotope

As mentioned earlier, the $\delta^{18}O_p$ were generated by a combination of bias correction and data fusion method for the 1870-2017 period. The CNN fusion was used for 1969-2007, and BCMs (mean of LS and DT) was used for the rest of the period.

Fig. 9 shows the monthly time series of generated $\delta^{18}O_p$ and their 12-month moving averages for eight sub-regions over the 1870-2017 period. TP is divided into monsoon and non-monsoon regions, according to the research of Yu et al. (2021). In our study, the region with significant correlation between $\delta^{18}O_p$ and temperature is the non-monsoon region, while the rest is the monsoon region. The Mann-Kendall tests show that the $\delta^{18}O_p$ significantly increased in NE and NC for recent 40 years at the P=0.01 level. These two regions are consistent with the temperature effect, and it can be inferred that the temperature of

these sub-regions has a rising trend during this period. This has been proved in many studies. For example, the studies of Ren et al. (2012) and Ding et al. (2007) have shown that the temperature in China has a rising trend in recent years, especially in northeast, north and northwest China. A slight upward trend is also observed in NW from the 1920s to the 1970s and 1980s to 2000s. The temperature effect is more significant in inland areas at middle and high latitudes. In winter, NW is mainly controlled by the westerlies, and the amount effect can be ignored (Yang et al., 2011; Yang et al., 2017).

Therefore, the temperature effect in NW is more significant in winter. From 2000 to 2012, the $\delta^{18}O_p$ values in NW showed a decreasing trend, mainly in winter. Some studies have shown that the temperature in NW during this period is consistent with the global land warming hiatus phenomenon, and even shows obvious cooling, especially in winter (Wen et al., 2017; Ma et al., 2019).

The $\delta^{18}O_p$ in SC presents a gentle upward trend in recent 70 years, indicating that precipitation has a downward trend in

this period, since the $\delta^{18}O_p$ conforms to the rule of amount effect in this region. While the $\delta^{18}O_p$ in SE shows a fluctuating trend of decline for the past 80 years (significant at 0.05 level), which indicates an upward trend in precipitation. These trends are consistent with the existing researches that precipitation increased in the east coast and northwest of China, but decreased in southwest, north and northeast (Zhai et al., 2005; Liang et al., 2016). There is no significant trend for $\delta^{18}O_p$ in TP. However, $\delta^{18}O_p$ shows a significant decreasing trend for the monsoon region of TP over the past 90 years, while it shows

a significant increasing trend for the non-monsoon region at the significant level of P=0.01. This is because the non-monsoon region shows temperature effect, while the monsoon region shows amount effect, which is consistent with the increasing



trend of temperature and precipitation in the Qinghai-Tibet region for recent years (Yu et al., 2021; Yao et al., 2013). Overall, the change trend of temperature and precipitation derived from the isotope effect analysis is consistent with that analysed directly using temperature and precipitation data in China. This further proved the reasonable performance of isoscape built in this study.

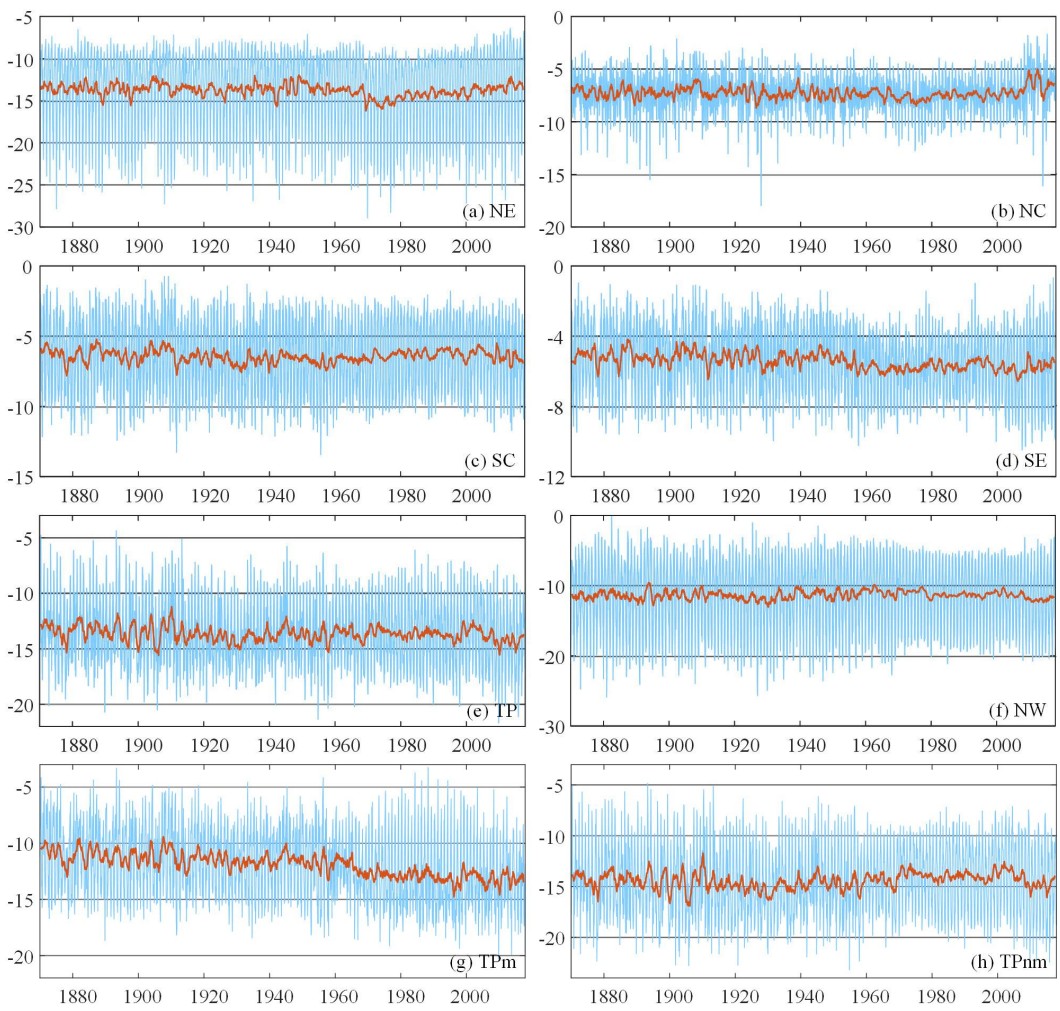

**Figure 9. Monthly time series of the generated δ¹⁸Oₚ (‰) and their 12-month moving average in eight sub-regions from 1870 to 2017. (g) shows the monsoon region. (h) shows the non-monsoon region.**

## 5 Data application

The dataset includes the stable oxygen isotope of precipitation for mainland China over the 1870-2017 period. Since the observation is limited and climate model simulations have different time durations, the entire duration was constructed by using the various methods and datasets for different periods as shown in Fig. 10.



| HadAM3 LS & DT ensemble average | CAM2 & HadAM3 LS & DT ensemble average | CAM2, GISS E & HadAM3 CNN fusion | CAM2, GISS E, HadAM3, LMDZ4 & MIROC32 CNN fusion | GISS E, LMDZ4 & MIROC32 CNN fusion | LMDZ4 zoomed LS & DT ensemble average |
|---|---|---|---|---|---|
| 1870-1957 | 1958-1968 | 1969-1978 | 1979-2001 | 2002-2007 | 2008-2017 |

**Figure 10. The generation mode of dataset in each period.**

Based on this new isoscape, the physical mechanisms driving the spatio-temporal variation of $\delta^{18}O_p$ can be deeply explored. This dataset is useful for tracing atmospheric and hydrological processes. For example, it can be used to study the effect of meteorological variables and air mass trajectory on stable isotope distribution, and quantify the source and fate of moisture (Tian et al., 2007; Peng et al., 2020b; Gao et al., 2011). The data can also help track the water cycle, to assess the source of water resources using complementary isotopic data from lakes, rivers and groundwater (Yao et al., 2013; Gao et al.,

2009). The generated dataset is also useful for paleoclimate studies, by studying the relationship between modern isotopes and meteorological variables (Yu et al., 2016b). When using this dataset, it should be noted that $\delta^{18}O_p$ for the 1979-2007 period generated using the CNN fusion method should be more reliable, because the performance comparison for BCMs and DFMs show that CNN fusion method performed better than BCMs, and more iGCM simulations were used for data fusion in this period that other periods.

**6 Data availability**

The generated dataset is available in Zenodo at https://doi.org/10.5281/zenodo.5703811 (Chen et al., 2021). The GNIP data can be obtained from the GNIP Database of IAEA/WMO (https://nucleus.iaea.org/wiser). The TNIP data was derived from the National Tibetan Plateau Data Centre (DOI: 10.11888/Geogra.tpdc.270940). The detailed information and sources of observations from other sources are shown in Table 3. The iGCM data can be downloaded from SWING2 (accessible at

https://data.giss.nasa.gov/swing2/), where LMDZ4 zoomed data was provided by Dr. Camille Risi at Laboratoire de Météorologie Dynamique in France.

**Table 3. Detailed information and sources of observations from previous publications in literatures.**

| Sub-region | Site | Latitude | Longitude | Period | Data source |
|---|---|---|---|---|---|
| SC | Heshang | 30.45 | 110.42 | 2011-2018 | Wang et al. (2020) |
| SC | Changsha | 28.25 | 112.55 | 2010-2017 | Zhou et al. (2019) (DOI: 10.17632/975k2wzw3p.1) |
| SE | Guangzhou | 23.13 | 113.32 | 2007-2014 | Yang et al. (2017) (DOI: 10.11888/AtmosPhys.tpe.249477.db.) |
| SE | Yongan | 28.89 | 120.85 | 2014-2018 | Hu et al. (2020) (DOI: 10.17632/vndfs3dpyn.1) |
| SE | Xishuangbanna | 21.93 | 101.27 | 2002-2004 | Liu et al. (2007) |



| Sub-region | Site | Latitude | Longitude | Period | Data source |
|---|---|---|---|---|---|
| SE | Guangzhou | 23.15 | 113.35 | 2007-2009 | Xie et al. (2011) |
| TP | Lulang | 29.77 | 94.73 | 2007-2014 | Yang et al. (2017) (DOI: 10.11888/AtmosPhys.tpe.249477.db.) |
| TP | Nuxia | 29.47 | 94.65 | 2009-2014 | Yang et al. (2017) (DOI: 10.11888/AtmosPhys.tpe.249477.db.) |
| TP | Yeniugou | 38.46 | 99.54 | 2008-2009 | Zhao et al. (2011) |

## 7 Conclusions

Long-time sequence of $\delta^{18}O_p$ are of great significance for hydrological and meteorological studies. In view of the lack of long and reliable $\delta^{18}O_p$ datasets in China, this study generates a new dataset by integrating multi-iGCM data to overcome the limitations of short duration and uneven distribution of observed data. This dataset contains monthly $\delta^{18}O_p$ over mainland China for the 1870-2017 period with a spatial resolution of 50-60 km. The dataset from 1969 to 2007 was generated by using the CNN fusion method, when the observed time series and multiple iGCM simulation are available. For other periods, it

was generated by bias correcting iGCMs simulations. Two BCMs (i.e. LS and DT) with similar performances were used to produce ensemble mean. Prior to building the isoscape, the performance of two BCMs (LS and DT) and three DFMs (BP, LSTM and CNN) were evaluated using RMSE and CC as criteria. The results showed that CNN fusion method consistently performed the best for all sub-regions in China, and BP and LSTM fusion methods performed slightly better than LS and DT (BCMs). The performance of LS and DT method was similar. In terms of spatial distribution and temporal variability of

$\delta^{18}O_p$, the generated data showed very similar spatial distributions to observations, and the temporal trend of $\delta^{18}O_p$ was consistent with the observed changes in precipitation and temperature for different regions in China. All these showed that the built isoscape is reliable and useful to extending the time and space of observations in China.

## Author contributions

JieC and XJZ designed the framework of the research. JiaC collected data, performed the analyses, and wrote the manuscript.

PP and CR provided data and methodological support. JieC supervised the progress of research and provided resources and financial support. All authors contributed to the discussion and revised the submitted manuscript.

## Competing interests

The authors declare that they have no conflict of interest.



**Acknowledgements**

We would like to thank the International Atomic Energy Agency (IAEA) for providing the GNIP data, the Chinese Academy of Sciences (CAS) for providing the CHNIP and TNIP data, and everyone who took samples and shared data. We would like to thank the Stable Water Isotope Intercomparison Group (SWING) and the second phase of SWING for providing the iGCM simulated data.

**Financial support**

This work was partially supported by the National Key Research and Development Program of China (grant no. 2017YFA0603704), the National Natural Science Foundation of China (grant nos. 52109007 and 52079093), the Overseas Expertise Introduction Project for Discipline Innovation (111 Project) funded by the Ministry of Education and State Administration of Foreign Experts Affairs, P. R. China (grant no. B18037).

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
