# Peer review of "A 148-year precipitation oxygen isoscape for China generated based on data fusion and bias correction of iGCMs simulations"

_Earth System Science Data, 2021_

## Author Comment (AC1)

**Responses to RC1:**

*The stable isotope composition of water is a very useful tracer to elucidate its formation history and cycle. The precipitation is an essential input for the hydrological cycle but it is very difficult to obtain continuous temporal and spatial variations of precipitation isotopes. This paper provides a high-resolution precipitation oxygen distribution in China based on IAEA long-term data and other datasets. These results will be of significant interest to the scientific community, particularly in hydrology. Overall, this review paper is scientifically robust given the available datasets. It is also reasonably written and well organized. I think the paper suits the reader of this journal and can be accepted for publication. I have some major comments and several minor comments for the authors to consider.*

**Re:** We thank the reviewer for the positive evaluation and valuable comments. We think all comments can be addressed in the revised manuscript. Our responses to each comment are presented as follows.

*Major:*

■ *The topic of this paper is very interesting, in my opinion, it upgrades the data from a sparse point scale to a continuous regional area scale. I suggest that authors highlight this in the introduction section, and be more explicit about the meaning of the paper, which will help arouse the reader's interest and facilitate its further spread.*

**Re:** Thanks for the suggestion. We agree with the reviewer that this paper upgrades the data from a sparse point scale to a continuous regional area scale. To improve our understanding of the precipitation and hydrological processes, it is necessary to have a long-term, temporally consistent and high-quality precipitation isoscape. Making full use of the advantages of observations and iGCM simulations, the built isoscape overcomes the deficiencies of the sparse and discontinuous distribution of observed data and reduces the uncertainty of iGCMs simulated data. We will modify the introduction to highlight the motivation of our study in the revised manuscript.

■ *As we know, isotopic composition in precipitation also varies dramatically over time. Therefore, I think the temporal resolution is also important. What do you think about this? I think authors should state or discuss clearly the time resolution.*

**Re:** We agree with the reviewer that the temporal resolution is very important for isoscape. The temporal resolution of isoscape provided in this study is at the monthly scale. In other words, twelve values are provided each year for each grid. The monthly temporal resolution was the most commonly used in scientific research for precipitation isotope composition, especially at the large spatial scale. Some observed event precipitation isotope data are also available, but they are

relatively short in duration and spatial scale. Thus, they were not considered in this study. The temporal resolution of isoscape will be clearly stated in the revised manuscript.

*Minor:*

*Line 25: investigated —> shown?*

**Re:** We will replace that.

*Line 40 add reference: IAEA/WMO (2022). Global Network of Isotopes in Precipitation. The GNIP Database. Accessible at: https://nucleus.iaea.org/wiser*

**Re:** We will add the reference.

*Line 63 watershed —> catchment*

**Re:** We will modify that.

*Figure 1 the unit (m) should be added to the legend.*

**Re:** The unit will be added in the figure.

*IAEA also provided long-term data in Haikou, why isn't it being used in your paper?*

**Re:** This is because the LMDZ4 zoomed data we obtained does not include Haikou. We will discuss this in the revised manuscript.

*Line 96 There are many types of averages, are you using a monthly precipitation amount weighted average here?*

**Re:** In most cases, we just use the monthly data provided by GNIP and CHNIP for our calculations. For a few references providing event isotope data, monthly precipitation weighted data are used. This will be clarified in the revised manuscript.

*Line 102 What is surface condition ï¼Ÿ would you like to give more information?*

**Re:** Yes. The free-running simulations were performed following the Atmospheric Model Intercomparison Project (AMIP) protocol, using prescribed sea surface temperatures (SST) and sea ice. We will clarified this and add the following references in the revised manuscript.

Risi, C., Bony, S., Vimeux, F., and Jouzel, J.: Water-stable isotopes in the LMDZ4 general circulation model: Model evaluation for present-day and past climates and applications to climatic interpretations of tropical isotopic records, Journal of Geophysical Research, 115, D12118, https://doi.org/10.1029/2009jd013255, 2010.

Yoshimura, K., Kanamitsu, M., Noone, D., and Oki, T.: Historical isotope simulation using Reanalysis atmospheric data, Journal of Geophysical Research, 113, D19108, https://doi.org/10.1029/2008jd010074, 2008.

*Line 219: Figure 3. I think the NW result is bad too. It should be discussed in detail.*

**Re:** We agree with the reviewer that the NW result is bad too. This is because, On the one hand, the sparse coverage of stations coupled with complex topography over northwest China cannot well represent the full range of precipitation isotope conditions. This can lead to biases in the distribution of observations. On the other hand, the arid northwest region is one of the most sensitive regions to climate change due to its fragile ecosystem, which affects sub-cloud evaporation and local moisture cycle, leading to the large uncertainty in isotope simulation between different iGCMs. This will be clarified in the revised manuscript.

Pang, Z., Kong, Y., Froehlich, K., Huang, T., Yuan, L., Li, Z., and Wang, F.: Processes affecting isotopes in precipitation of an arid region, Tellus B: Chemical and Physical Meteorology, 63, 352-359, https://doi.org/10.1111/j.1600-0889.2011.00532.x, 2011.

Wang, S., Zhang, M., Chen, F., Che, Y., Du, M., and Liu, Y.: Comparison of GCM-simulated isotopic compositions of precipitation in arid central Asia, Journal of Geographical Sciences, 25, 771-783, https://doi.org/10.1007/s11442-015-1201-z, 2015.

*Figure 5 – 8 What are SPR, SUM, AUT, and WIN? Jan- Mar is SPR? Authors should clearly indicate the months covered by each season. Table 2 also should be clearly mentioned.*

**Re:** SPR, SUM, AUT, and WIN are respectively defined as March – May (MAM), June – August (JJA), September – November (SON), and December – February (DJF). We will replace them with accurate definitions and add descriptions accordingly.

*Line 315-319. Author should add some references to prove your point.*

**Re:** The following references will be added in the revised manuscript.

Peng, D., and Zhou, T.: Why was the arid and semiarid northwest China getting wetter in the recent decades?, Journal of Geophysical Research: Atmospheres, 122, 9060-9075, https://doi.org/10.1002/2016JD026424, 2017.

Ren, G. Y., Yuan, Y. J., Liu, Y. J., Ren, Y. Y., Wang, T., and Ren, X. Y.: Changes in precipitation over Northwest China. Arid Zone Research, 33, 1-19, https://doi.org/10.13866/j.azr.2016.01.01, 2016.

Sun, C., Li, X., Chen, Y., Li, W., Stotler, R. L., and Zhang, Y.: Spatial and temporal characteristics of stable isotopes in the Tarim River Basin. Isotopes in Environmental and Health Studies, 52, 281-297, https://doi.org/10.1080/10256016.2016.1125350, 2016.

Yao, J., Chen, Y., Zhao, Y., Guan, X., Mao, W., and Yang, L.: Climatic and associated atmospheric water cycle changes over the Xinjiang, China, Journal of Hydrology, 585, 124823, https://doi.org/10.1016/j.jhydrol.2020.124823, 2020.

*Line338-40: You discussed TP again, why not move to Line302?*

**Re:** This is because in the first part (line 288-319), the spatial distribution of oxygen isotope in precipitation is analyzed, and after line 320, the seasonal variation of the isotope is analyzed.

*Line346: It's better to start a new paragraph so that it can be read more clearly.*

**Re:** Agree. We will modify the paragraph structure.

*Line 352-354: It's hard to see a significant trend.*

**Re:** This is because the trend of precipitation isotope in NW sub-region is not obvious. We have performed the Mann-Kendall test, which shows that δ18Op in NW increased significantly from the 1920s to the 1970s at the P=0.1 level. However, from the 1980s to the 2000s, the δ18Op shows an insignificant upward trend. This will be clarified in the revised manuscript.

*Figure 9 Need to add y-axis labels.*

**Re:** We will add the labels.

*Figure 10 I think it's very unclear to discuss why it is divided into these stages according to Figure 10. Based on methods? Or data? Authors should state it clearly.*

**Re:** This division is mainly based on the time period of iGCM data and performance of different fusion and bias correction methods. According to Section 4.1, CNN performs the best over all other fusion methods and bias correction methods, and the participation of more iGCMs in the CNN fusion further improve its performance. Therefore, when generating the isoscape, the CNN fusion was used in priority with the number of iGCM simulations being more than two. Considering that iGCMs have different time periods, we divide the whole period into various stages. CNN method was used in stages with sufficient iGCMs, and the bias correction method was used in other stages. This will be clarified in the revised manuscript.

---

## Author Comment (AC2)

**Responses to CC1:**

*The authors create a high-resolution precipitation oxygen isoscape dataset for China by fusing eight iGCMs simulations and in-situ observations based on data fusion and bias correction techniques. I appreciate the authors' great efforts to develop such a dataset, but I am quite dubious about the general content, and have large concerns about the novelty and quality of the dataset.*

**Re:** We would like to thank the reviewer for the time taking in reviewing our manuscript and providing constructive comments. Following these suggestions, we have carefully revised the manuscript, especially for the validation of the dataset's quality, adding more details and enriching the results. Below please find our point-by-point responses to these comments.

*My main concerns are:*
■ *Data-quality and novelty:*
*It feels like a direct comparison of five commonly used fusion methods for developing a high-resolution dataset in China, without any new advanced fusion methods. Furthermore, the quality of the developed dataset is still questionable and unreliable due to its poor and insufficient present form. It seems to me that the intended novelty might be a high-resolution dataset.*

**Re:** We would like to thank the reviewer for the insightful comments. We agree with the reviewer that there is indeed no new fusion method developed in this study. However, the main objective of this study is not to develop a new fusion method, but to use an appropriate combination of multiple methods to develop a high-resolution isoscape with a long time period for the region of China, since the observations and climate model simulations have various time periods and lengths. Based on the availability of observed and simulated dataset, multiple suitable methods were first compared and the most appropriate ones were used for each time periods This is a first attempt using a hybrid/mixed approach to develop such a dataset in China, which is the best existing dataset so far in the country and is of great importance in providing a data foundation for the study of complex hydrological and climatic systems.

The fusion and bias correction methods in the study have been widely used, even though they were not used in the field of stable isotopes. This study first compared the performance of these methods to find the best combination to build the dataset. The results showed that the CNN fusion method consistently performed the best for all sub-regions in China, and two bias correction methods (LS and DT) showed similar performance. The combination of the CNN fusion and bias correction methods is satisfactory to develop a high-resolution isoscape with a long time period. In other words, in order to maximize the utilization of available data, the CNN fusion method was used for the common period of all climate simulation and observations, while the bias correction methods were used for the periods with only one or two climate simulations, and with no observations. Considering China is a large country with various climatic conditions and complex terrain, the selected methods may also be able to be applied to other regions of the world.

Definitely, we agree with the reviewer that the dataset quality is the primary concern, and the quality was not meticulously presented in the original manuscript. In the revision of this manuscript, following the reviewers' specific comments, a comprehensive assessment of the dataset at a finer scale was conducted. Specifically, the dataset was evaluated with respect to reproducing the time-series of in-situ observations at station scale, as well as the spatial pattern for each month. The results show that the in-situ observations were reasonably represented by the isoscape time series for all stations. The correlation coefficients between the

isoscape simulations and observations are larger than 0.8 for 73% of the stations and larger than 0.9 for 49% of the stations. The root mean square errors between the isoscape simulations and observations are smaller than 20‰ for 78% of the stations and less than 15‰ for 56% of the stations. From the monthly spatial distribution of isotopes, CNN fusion simulations also capture the spatial pattern of observations most accurately. All results showed that the isoscape dataset is of high quality. **The detailed results can be found in the responses to specific comments below.**

*The methods used to develop the dataset are unclear and not robust. For example, the sensitivity of model parameters are not evaluated and discussed; thus, the results are not robust. Why does CNN perform much better? Why is it set to a three-layer structure model? The observed data covers a short period and is not sufficient to train the model.*

**Re:** We agree with the reviewer that the description of methods is not clear enough in the original manuscript. With strong abilities for generalization and information synthesis, neural networks can well coordinate various input information and deal with complex nonlinear relations (Hsu et al., 1995; Oyebode and Stretch, 2019). Then, they have been widely used in various fields (Reichstein et al., 2019). In particular, CNN performs exceptionally well because it has the ability to extract local features, and can grasp local correlation and spatial invariance well. These have also been proved in many literatures (Sadeghi et al., 2020; Jiang et al., 2021).

The structure and parameters of these neural network methods were carefully considered and validated. Considering that previous studies (e.g. Zhang and Wallace, 2015; Taylor et al., 2021; Mboga et al., 2017; Bengio, 2012) on parameter sensitivity of neural networks have shown similar results, we determined the parameter selection scheme based on these studies and used a hierarchical stepwise search method to determine parameter values. Specifically, the parameters were divided into three parts, structural parameters, sensitive algorithm parameters and other algorithm parameters, and then determined step by step. At each step, we tested the performance of all parameter combinations using the grid search method. Referring to previous studies (e.g. Xue et al., 2021; Wu et al., 2020; Langford et al., 2017; Chen et al., 2022), we considered some conventional parameter settings (such as filters are usually set to the power of 2). The details of parameter selection are shown in Table 1 (taking CNN as an example). Furthermore, when structural parameter values produced similar performance, we chose the simpler structure (i.e., the one with fewer parameters) to avoid overfitting. When algorithm parameter values produced similar performance, we chose the one that is more efficient for computing.

Table 1. Details of the parameter selection in Convolutional Neural Network (CNN).

| Steps | Parameters | Range tested | Selected |
|---|---|---|---|
| Step 1. Structural parameters | Convolutional layers | 1, 2, 3 | 2 |
| | Dense layers | 1, 2 | 1 |
| Step 2. Sensitive algorithm parameters | Learning rate | 0.0001-0.005 | 0.001 |
| | Batch size | 10-50 | 50 |
| | Filter size | 3, 4, 5 | 3 |
| | Filters | 8, 16, 32 | 8/32 |
| Step 3. Other algorithm parameters | Dense neurons | 8, 16, 32 | 32 |
| | Dropout rate | 0.1-0.3 | 0.1 |
| | Activation | ReLU, TanH | ReLU |

We also agree with the reviewer that the time period of observed data is short for some stations. However, the monthly data were used in our study. More importantly, to partly solve the problem of short period, the data fusion and bias correction were conducted at the regional scale for a specific season to include several stations for training the neural network models and bias correction methods. In other words, the training of model was not conducted for each individual station. This ensures that the model was well trained with enough samples. Moreover, considering the lack of observed data for training, we chose the simple structure to make the network not deep and the results desirable. For example, we chose the 1D convolutional neural network because of its advantages for data scarcity (Kiranyaz et al., 2021).

Overall, **the evaluation of the isoscape dataset show that the dataset quality is excellent (Figs. 1-2 and Tables 2-3)**, further proving the structure and parameters of the CNN fusion method are appropriate. Definitely, the models might perform even better if more observations were included. But then again, the lack of observations is the main motivation to develop this dataset to fill the data gaps both in space and time, as we were attempted to maximize the utilization of available data from all observations and climate model simulations to provide best available dataset to date.

**All above information will be added in the discussion of the revised manuscript.**

References:

Bengio, Y.: Practical recommendations for gradient-based training of deep architectures, Neural networks: Tricks of the trade, Springer, Berlin, Heidelberg, 437-478, https://doi.org/10.1007/978-3-642-35289-8_26, 2012.

Chen, H., Sun, L., Cifelli, R., and Xie, P.: Deep Learning for Bias Correction of Satellite Retrievals of Orographic Precipitation, IEEE Transactions on Geoscience and Remote Sensing, 60, 1-11, https://doi.org/10.1109/TGRS.2021.3105438, 2022.

Hsu, K.-l., Gupta, H. V., and Sorooshian, S.: Artificial Neural Network Modeling of the Rainfall-Runoff Process, 31, 2517-2530, https://doi.org/10.1029/95WR01955, 1995.

Jiang, Y., Yang, K., Shao, C., Zhou, X., Zhao, L., Chen, Y., and Wu, H.: A downscaling approach for constructing high-resolution precipitation dataset over the Tibetan Plateau from ERA5 reanalysis, Atmospheric Research, 256, 105574, https://doi.org/10.1016/j.atmosres.2021.105574, 2021.

Kiranyaz, S., Avci, O., Abdeljaber, O., Ince, T., Gabbouj, M., and Inman, D. J.: 1D convolutional neural networks and applications: A survey, Mechanical systems and signal processing, 151, 107398, https://doi.org/10.1016/j.ymssp.2020.107398, 2021.

Langford, Z. L., Kumar, J., and Hoffman, F. M.: Convolutional neural network approach for mapping arctic vegetation using multi-sensor remote sensing fusion, 2017 IEEE International Conference on Data Mining Workshops (ICDMW), 322-331, https://doi.org/10.1109/ICDMW.2017.48, 2017.

Mboga, N., Persello, C., Bergado, J. R., and Stein, A.: Detection of informal settlements from VHR images using convolutional neural networks, Remote sensing, 9, 1106, https://doi.org/10.3390/rs9111106, 2017.

Oyebode, O., and Stretch, D.: Neural network modeling of hydrological systems: A review of implementation techniques, Natural Resource Modeling, 32, e12189, https://doi.org/10.1111/nrm.12189, 2019.

Reichstein, M., Camps-Valls, G., Stevens, B., Jung, M., Denzler, J., and Carvalhais, N.: Deep learning and process understanding for data-driven Earth system science, Nature, 566, 195-204, https://doi.org/10.1038/s41586-019-0912-1, 2019.

Sadeghi, M., Nguyen, P., Hsu, K., and Sorooshian, S.: Improving near real-time precipitation estimation using a U-Net convolutional neural network and geographical information, Environmental Modelling & Software, 134, 104856, https://doi.org/10.1016/j.envsoft.2020.104856, 2020.

Taylor, R., Ojha, V., Martino, I., and Nicosia, G.: Sensitivity analysis for deep learning: ranking hyper-parameter influence,

2021 IEEE 33rd International Conference on Tools with Artificial Intelligence (ICTAI), 512-516, https://doi.org/10.1109/ICTAI52525.2021.00083, 2021.

Wu, H., Yang, Q., Liu, J., and Wang, G.: A spatiotemporal deep fusion model for merging satellite and gauge precipitation in China, Journal of Hydrology, 584, 124664, https://doi.org/10.1016/j.jhydrol.2020.124664, 2020.

Xue, M., Hang, R., Liu, Q., Yuan, X. T., and Lu, X.: CNN-based near-real-time precipitation estimation from Fengyun-2 satellite over Xinjiang, China, Atmospheric Research, 250, 105337, https://doi.org/10.1016/j.atmosres.2020.105337, 2021.

Zhang, Y., and Wallace, B.: A sensitivity analysis of (and practitioners' guide to) convolutional neural networks for sentence classification, arXiv [preprint], arXiv:1510.03820, 2015.

*Using the interpretations of spatial pattern at the seasonally averaged scale (1969-2007) and the temporal pattern at the regional scale to validate the effectiveness and reliability of the data is not persuasive. Why not give us a comprehensive assessment at finer scale, such as time-series comparisons between the gridded simulations (50km) and in-situ observations at each station, and spatial patterns for each month. Without these comprehensive evaluations at finer scale, I am quite dubious about the data-quality and usefulness of this data set.*

**Re:** Thanks lot for the insightful suggestions. **Following the reviewer's comments, a comprehensive assessment of the dataset at the finer scale was conducted and included in the revision.** Specifically, the dataset was evaluated with respect to reproducing the time-series of in-situ observations at station scale, as well as the spatial pattern for each month. The detailed procedures are presented as follows.

To evaluate the dataset for all stations, the correlation coefficient (CC) and root mean square error (RMSE) were calculated for $\delta^{18}O_p$ series between observations and raw iGCM simulations, and between observations and built isoscape for all stations over the common period (Tables 2-3). The results show that the built isoscape performs excellent for the vast majority of stations, with larger CCs and smaller RMSEs than iGCM simulations. Specifically, the CCs between the isoscape simulations and observations are larger than 0.8 for 73% of the stations and larger than 0.9 for 49% of the stations. The RMSEs between the isoscape simulations and observations are smaller than 20‰ for 78% of the stations and less than 15‰ for 56% of the stations.

To further demonstrate the dataset quality, two stations with appropriate length of observation were randomly selected for each sub-region. Totally, 12 stations were selected. The time series of $\delta^{18}O_p$ were plotted for observations, iGCM simulations, and the generated isoscape (Fig. 1). As can be seen from Fig. 1, the variations of $\delta^{18}O_p$ are very consistent between the generated isoscape and observations, and the isoscape performs much better than raw iGCM simulations. In particular for the period before 2007, the CNN model integrates the advantages of various simulations and captures most features of the observed data. These results generally prove that the generated isoscape is reliable.

Fig. 2 further shows the monthly spatial distribution of observed, newly generated, and better-performing simulated $\delta^{18}O_p$ for their common period (i.e. 1979-2007). The spatial pattern presented by the built isoscape shows the best consistency with the observations. The strength of the CNN model has been demonstrated, which can make good use of the advantages of each simulation to accurately capture the characteristics of observations. For example, the LMDZ nudged model shows a strong ability to reproduce the spatial distribution of $\delta^{18}O_p$ for the eastern region in summer and autumn, but a slightly poor performance in the Qinghai-Tibet Plateau. The built isoscape combines LMDZ nudged with GISS nudged and LMDZ zoomed simulations, which show reasonably performance for the Qinghai-Tibet Plateau, and well reproduces the spatial distribution of $\delta^{18}O_p$ for mainland China.

In addition, based on suggestions from other reviewers, we will introduce some physical-based ancillary data in the fusion, such as elevation and meteorological data, to refine our dataset. Taking into account the effects of elevation and meteorological factors on precipitation isotopes might make our dataset more reliable.

**From above analyzes, we are very confident that the generated isoscape is of high quality, and will be widely used in the future. All above results will be added in the revised manuscript.**

■ *The presentation of dataset.*
*Introduction: The section of Introduction is not well written, and lacks to interconnect of the data it shares to and to show how it is valuable in relation to the Earth's system. For example, readers should have a clear understanding of the motivation of this study, the purpose of creating such a dataset, which can be seen from a literature review of precipitation oxygen isoscape in hydrological and biogeochemical cycles. Then, followed by a detailed description of the available datasets, we could not find any description of the previously evaluated performance of iGCMs simulation in the Introduction. A description of the data-fusion method should also be added.*

**Re:** We agree with the reviewer that the Introduction has a large room to be improved. We will revise the introduction as suggested, and some of the following information will be added in the Introduction of the revised manuscript.

"Taking advantage of the fact that isotope composition varies sensitively with environmental conditions, environmental isotopes play an important role in the identification and characterization of the Earth's systems processes (Bowen, 2010). The study of the hydrologic cycle is one of the most important applications of stable isotopes. Firstly, the isotope composition of water is a powerful tracer of water sources. In the process of moisture transport, the isotope composition changes with atmospheric processes, which can reflect moisture contribution (Gibson et al., 2005; Galewsky et al., 2016; Ansari et al., 2020). In addition, for surface runoff, soil water and groundwater, the isotope composition can also reflect the water source, infiltration mechanism and evaporation consumption of each system (Zhang et al., 2021; McGuire et al., 2002; Gazis and Feng, 2004; Chen et al., 2004). Secondly, isotope composition can also reveal the hydrological processes that cannot be achieved by other methods (Bowen et al., 2019). For example, evaporation processes can be better diagnosed in dual hydrogen/oxygen (H/O) or triple oxygen isotope ($^{18}O$-$^{17}O$-$^{16}O$) datasets, which can be used to quantify the effect of raindrop re-evaporation on atmospheric water balance (Worden et al., 2007; Froehlich et al., 2008). Thirdly, isotopes can be incorporated into surface hydrology models as diagnostic tools. The isotope composition of evapotranspiration, soil moisture, and runoff can be predicted by incorporating the isotope cycle, thus the distribution of isotopic variation in evapotranspiration and runoff can be better understood (Fekete et al., 2006). What's more, isotope composition can quantify evaporation rates, which is useful for understanding water balance and climate change from catchment to continental scales (Bowen, 2010). Precipitation isotopes can also be used to estimate the precipitation isotopic lapse rate by establishing relationships with climatic elements or elevation, so as to study paleoclimate and paleoelevation based on isotopes (Rowley and Garzione, 2007; Johnson and Ingram, 2004).

The international observation of stable isotopes in precipitation began in the 1950s, and the Global Network of Isotopes in Precipitation (GNIP) established in 1961 provide first-hand data for the study of stable isotopes in precipitation. Since then, Austria (Austrian Network of Isotopes in Precipitation, ANIP, Kralik et al., 2003), the United States (United State Network of Isotopes in Precipitation, USNIP, Lynch et al., 1995), Switzerland (Swiss National Network for the Observation of Isotopes in the Water Cycle, NISOT, Schürch et al., 2003), Canada (Canadian Network of Isotopes in Precipitation, CNIP, Frits et al., 1987) and

other countries have also established their national networks, which provide strong data support for promoting and deepening the study of stable precipitation isotope.

The establishment of the isotope observation network in China was relatively late. Before 1985, GNIP had only one station in Hong Kong of China, and it was not until 1985 that more stations were selected for inclusion in GNIP. Due to the scarcity of stations on the Tibetan Plateau, the Chinese Academy of Sciences (CAS) launched the Tibetan Plateau Network of Isotopes in Precipitation (TNIP) in 1991 (Yu et al., 2016). However, most Chinese stations in GNIP stopped monitoring in the early 2000s (Zhang and Wang, 2016), and until 2004 only one station remained. In order to continue the systematic study, the CAS established the Chinese Network of Isotopes in Precipitation (CHNIP) based on the Chinese Ecosystem Research Network (CERN) in 2004 (Song et al., 2007).

The stable isotopes in precipitation can also be simulated by isotope-equipped general circulation models (iGCMs). In contrast to observations, iGCMs can provide time-continuous and space-regular isotope data. Joussaume et al. (1984) incorporated the fractionation process of water stable isotope into GCM for the first time. They used the GCM of the Laboratoire de Météorologie Dynamique (LMD) to simulate the distribution of global water stable isotope, and the relationship between simulated precipitation oxygen isotope and meteorological elements was in good agreement with the measured results. Since then, an increasing number of GCMs have incorporated isotope cycles, for example, the ECHAM4 developed by the Max Planck Institute for Meteorology (MPI) in Germany, the GISS E developed by the NASA Goddard Institute for Space Studies (GISS) in the United States, the HadAM3 developed by the Hadley Centre for Climate Prediction and Research in the United Kingdom, the LMDZ4 developed by the Laboratoire de Météorologie Dynamique in France, and the MIROC32 developed by the Center for Climate System Research (CCSR) of the University of Tokyo in Japan, etc.

On the basis of these, the comparison and evaluation of iGCMs in simulating isotopes have been conducted in many studies. Yoshimura et al. (2003) indicated that due to the limitation of spatial and temporal resolution, iGCMs are poor in simulating the short-term (days) variability of stable isotopes in precipitation, while they are good at the monthly or annual scale. Conroy et al. (2013) evaluated the spatio-temporal pattern of precipitation isotope variability in the tropical Pacific for iGCM simulations, and found that models nudged by reanalysis wind have a certain effect on precipitation isotope values, and the performance of models varies with regions. Zhang et al. (2012) selected four iGCMs to evaluate the average precipitation isotopic composition in East Asia. The results showed that the characteristics of measured values were well reproduced by iGCM simulation, but the simulated values were all lower in the inland areas at middle and high latitudes, and the amount effect in arid areas was incorrectly simulated. Wang et al. (2015) verified iGCM-simulated stable isotopes in precipitation in arid Central Asia. In general, the seasonality of stable isotopes in precipitation could be well simulated, but the values of oxygen isotopes were higher in summer and lower in winter, lower in the eastern section and higher in the western section. Che et al. (2016) concluded that LMDZ nudged has the best comprehensive performance by comparing the simulated values of different models with the measured values of GNIP in China. In terms of altitude effects, CAM and GISS E perform better, while in terms of continental effects, GISS E and LMDZ free performs better.

To comprehensively consider the error characteristics and advantages of different sources of data to reduce the uncertainty, data fusion is usually used. One of the common methods for data fusion is to use in-situ observations as baselines to correct estimates from other sources. Several data fusion methods such as cokriging (Krajewski, 1987), probability matching (Rosenfeld et al., 1994), statistical objective analysis (Pereira et al., 1998), Bayesian correction (Todini, 2001), probability density function–optimal interpolation (Shen et al., 2014), and variational (Bianchi et al., 2013) are usually used to fuse in-situ observation

information. The key of these methods is to deal with the estimation errors directly based on weighted average, regression analysis, filtering analysis and other mathematical approaches. In contrast, neural network methods have stronger learning and generalization abilities, and have advantages in discovering complex relationships in data and processing large amounts of data (LeCun et al., 2015). So far, the neural network was mainly applied to precipitation data fusion in the field of hydrology but very little in isotopic hydrology. For example, Turlapaty et al. (2010) used Artificial Neural Network to fuse various satellite precipitation products, and found the fusion performance was statistically superior to each individual dataset for all seasons. Sun and Tang (2020) combined information from satellite precipitation products and reanalysis data in Central Texas, U.S., by using an attention-based deep convolutional neural network (AU-Net), and found the Au-net models have achieved varying degrees of success under different climatic conditions. Wu et al. (2020) combined Convolutional Neural Network with Long Short-Term Memory Network to fuse the TRMM satellite data, thermal infrared images of Gridded satellite, rain gauge data and elevation data. The results showed that this method can improve the accuracy of original TRMM data in China, even for regions with different precipitation intensities or sparse gauges."

References:

Ansari, M. A., Noble, J., Deodhar, A., and Saravana Kumar, U.: Atmospheric factors controlling the stable isotopes ($\delta 18O$ and $\delta 2H$) of the Indian summer monsoon precipitation in a drying region of Eastern India, Journal of Hydrology, 584, 124636, https://doi.org/10.1016/j.jhydrol.2020.124636, 2020.

Baez-Villanueva, O. M., Zambrano-Bigiarini, M., Beck, H. E., McNamara, I., Ribbe, L., Nauditt, A., and Thinh, N. X.: RF-MEP: A novel Random Forest method for merging gridded precipitation products and ground-based measurements, Remote Sensing of Environment, 239, 111606, https://doi.org/10.1016/j.rse.2019.111606, 2020.

Bianchi, B., Jan van Leeuwen, P., Hogan, R. J., and Berne, A.: A variational approach to retrieve rain rate by combining information from rain gauges, radars, and microwave links, Journal of Hydrometeorology, 14, 1897-1909, https://doi.org/10.1175/JHM-D-12-094.1, 2013.

Bowen, G. J.: Isoscapes: spatial pattern in isotopic biogeochemistry, Annual review of earth and planetary sciences, 38, 161-187, https://doi.org/10.1146/annurev-earth-040809-152429, 2010.

Bowen, G. J., Cai, Z., Fiorella, R. P., and Putman, A. L.: Isotopes in the water cycle: regional-to global-scale patterns and applications, Annual Review of Earth and Planetary Sciences, 47, 453-479, https://doi.org/10.1146/annurev-earth-053018-060220, 2019.

Che, Y., Zhang, M., Wang, S., Wang, J., Liu, Y., and Zhang, F.: Stable water isotopes of precipitation in China simulated by SWING2 models, Arabian Journal of Geosciences, 9, 1-12, https://doi.org/10.1007/s12517-016-2755-5, 2016.

Chen, J. S., Li, L., Wang, J. Y., Barry, D. A., Sheng, X. F., Gu, W. Z., Zhao, X., and Chen, L.: Groundwater maintains dune landscape, Nature, 432, 459-460, https://doi.org/10.1038/432459a, 2004.

Conroy, J. L., Cobb, K. M., and Noone, D.: Comparison of precipitation isotope variability across the tropical Pacific in observations and SWING2 model simulations, Journal of Geophysical Research: Atmospheres, 118, 5867-5892, https://doi.org/10.1002/jgrd.50412, 2013.

Fekete, B. M., Gibson, J. J., Aggarwal, P., and Vörösmarty, C. J.: Application of isotope tracers in continental scale hydrological modeling, Journal of Hydrology, 330, 444-456, https://doi.org/10.1016/j.jhydrol.2006.04.029, 2006.

Frits, P., Drimmie, R. J., Frape, S. K., and O'shea, K.: The isotopic composition of precipitation and groundwater in Canada, International symposium on the use of isotope techniques in water resources development, Vienna, Austria, 30 Mar - 3 Apr 1987, 1987.

Froehlich, K., Kralik, M., Papesch, W., Rank, D., Scheifinger, H., and Stichler, W.: Deuterium excess in precipitation of Alpine regions–moisture recycling, Isotopes in Environmental and Health Studies, 44, 61-70,

https://doi.org/10.1080/10256010801887208, 2008.

Galewsky, J., Steen-Larsen, H. C., Field, R. D., Worden, J., Risi, C., and Schneider, M.: Stable isotopes in atmospheric water vapor and applications to the hydrologic cycle, Rev Geophys, 54, 809-865, https://doi.org/10.1002/2015RG000512, 2016.

Gazis, C., and Feng, X.: A stable isotope study of soil water: evidence for mixing and preferential flow paths, Geoderma, 119, 97-111, https://doi.org/10.1016/S0016-7061(03)00243-X, 2004.

Gibson, J. J., Edwards, T. W. D., Birks, S. J., St Amour, N. A., Buhay, W. M., McEachern, P., Wolfe, B. B., and Peters, D. L.: Progress in isotope tracer hydrology in Canada, Hydrological Processes, 19, 303-327, https://doi.org/10.1002/hyp.5766, 2005.

Johnson, K. R., and Ingram, B. L.: Spatial and temporal variability in the stable isotope systematics of modern precipitation in China: implications for paleoclimate reconstructions, Earth and Planetary Science Letters, 220, 365-377, https://doi.org/10.1016/S0012-821X(04)00036-6, 2004.

Joussaume, S., Sadourny, R., and Jouzel, J.: A general circulation model of water isotope cycles in the atmosphere, Nature, 311, 24-29, https://doi.org/10.1038/311024a0, 1984.

Krajewski, W. F.: Cokriging radar-rainfall and rain gage data, Journal of Geophysical Research: Atmospheres, 92, 9571-9580, https://doi.org/10.1029/JD092iD08p09571, 1987.

Kralik, M., Papesch, W., and Stichler, W.: Austrian Network of Isotopes in Precipitation (ANIP): Quality assurance and climatological phenomenon in one of the oldest and densest networks in the world, International symposium on isotope hydrology and integrated water resources management, Vienna, Austria, 19-23 May 2003, 146-149, 2003.

LeCun, Y., Bengio, Y., and Hinton, G.: Deep learning, Nature, 521, 436-444, https://doi.org/10.1038/nature14539, 2015.

Lynch, J. A., Grimm, J. W., and Bowersox, V. C.: Trends in precipitation chemistry in the United States: A national perspective, 1980–1992, Atmospheric Environment, 29, 1231-1246, https://doi.org/10.1016/1352-2310(94)00371-Q, 1995.

McGuire, K. J., DeWalle, D. R., and Gburek, W. J.: Evaluation of mean residence time in subsurface waters using oxygen-18 fluctuations during drought conditions in the mid-Appalachians, Journal of Hydrology, 261, 132-149, https://doi.org/10.1016/S0022-1694(02)00006-9, 2002.

Pereira Fo, A. J., Crawford, K. C., and Hartzell, C. L.: Improving WSR-88D hourly rainfall estimates, Weather and Forecasting, 13, 1016-1028, https://doi.org/10.1175/1520-0434(1998)013<1016:IWHRE>2.0.CO;2, 1998.

Rosenfeld, D., Wolff, D. B., and Amitai, E.: The window probability matching method for rainfall measurements with radar, Journal of Applied Meteorology and Climatology, 33, 682-693, https://doi.org/10.1175/1520-0450(1994)033<0682:TWPMMF>2.0.CO;2, 1994.

Rowley, D. B., and Garzione, C. N.: Stable isotope-based paleoaltimetry, Annual Review of Earth and Planetary Sciences, 35, 463-508, https://doi.org/10.1146/annurev.earth.35.031306.140155, 2007.

Schürch, M., Kozel, R., Schotterer, U., and Tripet, J. P.: Observation of isotopes in the water cycle—the Swiss National Network (NISOT), Environmental Geology, 45, 1-11, https://doi.org/10.1007/s00254-003-0843-9, 2003.

Shen, Y., Zhao, P., Pan, Y., and Yu, J.: A high spatiotemporal gauge-satellite merged precipitation analysis over China, Journal of Geophysical Research: Atmospheres, 119, 3063-3075, https://doi.org/10.1002/2013JD020686, 2014.

Song, X. F., Liu, J. R., Sun, X. M., Yuan, G. F., Liu, X., and Wang, S. Q.: Establishment of Chinese Network of Isotopes in Precipitation (CHNIP) Based on CERN, Advances in Earth Science, 22, 738, https://doi.org/10.11867/j.issn.1001-8166.2007.07.0738, 2007.

Sun, A. Y., and Tang, G.: Downscaling satellite and reanalysis precipitation products using attention-based deep convolutional neural nets, Frontiers in Water, 56, https://doi.org/10.3389/frwa.2020.536743, 2020.

Turlapaty, A. C., Anantharaj, V. G., Younan, N. H., and Turk, F. J.: Precipitation data fusion using vector space transformation and artificial neural networks, Pattern Recognition Letters, 31, 1184-1200, https://doi.org/10.1016/j.patrec.2009.12.033, 2010.

Todini, E.: A Bayesian technique for conditioning radar precipitation estimates to rain-gauge measurements, Hydrology and

Earth System Sciences, 5, 187–199, https://doi.org/10.5194/hess-5-187-2001, 2001.

Wang, S., Zhang, M., Chen, F., Che, Y., Du, M., and Liu, Y.: Comparison of GCM-simulated isotopic compositions of precipitation in arid central Asia, Journal of Geographical Sciences, 25, 771-783, https://doi.org/10.1007/s11442-015-1201-z, 2015.

Worden, J., Noone, D., and Bowman, K.: Importance of rain evaporation and continental convection in the tropical water cycle, Nature, 445, 528-532, https://doi.org/10.1038/nature05508, 2007.

Wu, H., Yang, Q., Liu, J., and Wang, G.: A spatiotemporal deep fusion model for merging satellite and gauge precipitation in China, Journal of Hydrology, 584, 124664, https://doi.org/10.1016/j.jhydrol.2020.124664, 2020.

Yoshimura, K., Oki, T., Ohte, N., and Kanae, S.: A quantitative analysis of short-term 18O variability with a Rayleigh-type isotope circulation model, Journal of Geophysical Research: Atmospheres, 108, https://doi.org/10.1029/2003JD003477, 2003.

Yu, W., Wei, F., Ma, Y., Liu, W., Zhang, Y., Luo, L., Tian, L., Xu, B., and Qu, D.: Stable isotope variations in precipitation over Deqin on the southeastern margin of the Tibetan Plateau during different seasons related to various meteorological factors and moisture sources, Atmospheric Research, 170, 123-130, https://doi.org/10.1016/j.atmosres.2015.11.013, 2016.

Zhang, M. and Wang, S.: A review of precipitation isotope studies in China: Basic pattern and hydrological process, Journal of Geographical Sciences, 26, 921-938, https://doi.org/10.1007/s11442-016-1307-y, 2016.

Zhang, X., Sun, Z., Guan, H., Zhang, X., Wu, H., and Huang, Y.: GCM simulations of stable isotopes in the water cycle in comparison with GNIP observations over East Asia, Acta Meteorologica Sinica, 26, 420-437, https://doi.org/10.1007/s13351-012-0403-x, 2012.

Zhang, Y., Jones, M., Zhang, J., McGowan, S., and Metcalfe, S.: Can δ18O help indicate the causes of recent lake area expansion on the western Tibetan Plateau? A case study from Aweng Co, Journal of Paleolimnology, 65, 169-180, https://doi.org/10.1007/s10933-020-00158-6, 2021.

*Data set and study area: Please add more details about the in-situ data (e.g., time-series of available in-situ data, number of data points at each station) in the supplementary material to justify the machine learning. For better visualization, a mesh of the iCGMs could be added in Fig.1.*

**Re:** As suggested, a table (Table 4) will be added in the supplementary material to describe the details of the in-situ data, and a mesh of the built isoscape will be added in Fig.1 in the revised manuscript.

*Methodology: This section is not well written, and should be clarified. For example, did you correct for iGCMs simulation bias at grid scale (grid by grid) or regional scale (using all gauges and all iGCMs within a specific region). How do these methods generate 50 km simulations from various iGCMs with different spatial resolutions?*

**Re:** Thanks for the detailed comments. Generally, the generation of isoscape can be divided into five steps. (1) Prior to generating the dataset, a simple but robust inverse distance weighting (IDW) method (Camera et al., 2014) was used to interpolate all iGCM simulations to observation stations. (2) Three neural network data fusion and two bias correction methods were trained using observations and iGCM simulations for all months within a season and all stations within a sub-region. In other words, observed and simulated monthly isotopes within a season and all stations within a region were used to train the data fusion and bias correction methods. This ensures that the model was well trained with enough samples. (3) The performance of each model was evaluated for the validation period by the cross-validation method to find the optimal data fusion and bias correction methods. (4) All iGCM simulations were interpolated to the LMDZ4 zoomed grid with a spatial

resolution of approximately 50 km by the IDW method. (5) The optimal trained model and bias correction methods were applied to all grid points within a region and all months within a season. Since the length of iGCM simulations is not identical, the optimal combination of data fusion and bias correction methods were used to generate the isoscape for a long period. In other words, for the common period of observations and iGCM simulations, the optimal data fusion method (i.e. CNN) was used, while for the period with no observations, the bias correction methods were used.

All above descriptions will be added in the revised manuscript.

References:

Camera, C., Bruggeman, A., Hadjinicolaou, P., Pashiardis, S., and Lange, M. A.: Evaluation of interpolation techniques for the creation of gridded daily precipitation (1× 1 km2); Cyprus, 1980–2010, Journal of Geophysical Research: Atmospheres, 119, 693-712, https://doi.org/10.1002/2013JD020611, 2014.

*Results and discussion: see above comments. Many published ESSD papers have demonstrated the uncertainties, limitations, advantages and 1-3 specific applications (for validation, evaluation, and analysis) of their dataset, which are also highly recommended for your paper.*

**Re:** Thanks for suggestions. As suggested, we will add all relevant information to the revised manuscript.

(1) Uncertainties

In the original manuscript, the uncertainty was partly analyzed, but not comprehensive. The plotted +/- one standard deviation in Figs. 5-6 shows the dispersion degree of the evaluation metrics (CC and RMSE) for the simulated results of all bias correction and data fusion methods over 100 trials. The standard deviations of CC and RMSE calculated by LS and DT corrected simulations are smaller, while those calculated by BP, LSTM and CNN fused simulations are larger. The standard deviation of CC and RMSE calculated by the CNN fused simulations is the smallest among fusion methods. It can be considered that LS and DT correction methods show smaller uncertainties in CC and RMSE than the BP, LSTM and CNN fusion methods. CNN fusion methods show smaller uncertainties than the other two fusion methods.

In the revised manuscript, an uncertainty analysis for the use of data fusion methods was further analyzed. Specifically, the CNN method was taken as an example to analyze the uncertainty derived from model structure, model parameters and training samples for fusing isotope in South China (SC) over summer (JJA). For the model structure, a different number of convolution layers were selected, because the model is very sensitive to the number of convolution layers (Mboga et al., 2017). For model parameters, three parameters, namely learning rate, batch size and filter size, were selected based on previous sensitivity studies (Zhang and Wallace, 2015; Taylor et al., 2021; Mboga et al., 2017; Bengio, 2012). Commonly-used values for each parameter were selected to form twelve groups of parameter setting schemes. For the training sample, five different training-test sets were randomly generated. To sum up, the modelling combination scheme for uncertainty analysis is shown in Table 5.

A variance decomposition method (Song et al., 2020; Bosshard et al., 2013) was used to calculate the uncertainty contribution for these three sources as well as their interactions. The correlation coefficient (CC) and root mean square error (RMSE) were used as the evaluation criteria. Then, all the combination schemes were trained and the evaluation criteria were calculated, which was repeated 30 times for each combination. Results shows that the accuracy (i.e. standard error) of CC and RMSE are respectively 0.0025 and 0.0127‰. The relative contribution of each source to the total uncertainty is shown in Fig. 3. As can be seen, there is little difference in the relative contribution of uncertainty sources between the two evaluation criteria. Model

parameters have the greatest contribution to the total uncertainty with the contribution being more than 50%, while the contribution of training samples is the least, which is less than 1%. These results indicate that the model is robust and not very dependent on training data.

Table 5. The modeling combination scheme of uncertainty calculation.

| Model structure | Model parameters | | | Training samples |
|---|---|---|---|---|
| Number of convolutional layer | Learning rate | Batch size | Filter size | |
| 1 | 0.0005 | 20 | 3 | |
| 2 | 0.001 | 50 | 4 | Samples 1-5 |
| 3 | 0.002 | | | |

[Figure]

Figure 3. The relative contribution of each source to the total uncertainty.

(2) Advantages and limitations

The generated isoscape dataset has high spatio-temporal resolution and a long series covering 1870-2017. Compared with the existing iGCMs, the isoscape has high quality and stability for a large region in China at the monthly scale. Benefiting from the characteristics of optimal neural network and bias correction methods, the isoscape makes full use of observations to integrate the advantages of various iGCMs. In other words, by using the combination of data fusion and bias correction methods, all observations and iGCM simulations were used to the utmost extent to ensure the highest accuracy throughout the entire time period. The uncertainty analysis show that the CNN model is not dependent on specific training samples and has a strong generalization ability, while the bias correction methods have commonly used in climate change studies. Moreover, the hybrid generation method of the isoscape has the characteristics of high accuracy and simplicity, which can be easily extended to the generation of isoscape datasets in other regions. However, it should be noted that the isoscape may be more reliable for the common periods of most iGCMs (1969-2007), but mediocre for other periods. What's more, affected by the data quality and representativeness of observation stations, the accuracy of the isoscape in some regions still needs to be improved. It is believed that this problem will be solved as observed data become more abundant.

(3) Applications

The isoscape is very useful for tracking moisture sources and quantifying moisture contributions. For example over East Asia, where the length of observed isotope data are short, or over the Tibetan Plateau, where data are unevenly distributed, the influence of climate change on moisture source and contribution can be studied based on this long series precipitation isoscape. The isoscape can also be used to calibrate climate models through data assimilation. For example, the precipitation isoscape can be combined with the physical constraints of regional climate models to reconstruct hydrological and climatic elements such as water vapor and precipitation. It can be a useful attempt to advance the study of climatic and hydrological data.

References:

Bengio, Y.: Practical recommendations for gradient-based training of deep architectures, Neural networks: Tricks of the trade, Springer, Berlin, Heidelberg, 437-478, https://doi.org/10.1007/978-3-642-35289-8_26, 2012.

Bosshard, T., Carambia, M., Goergen, K., Kotlarski, S., Krahe, P., Zappa, M., and Schär, C.: Quantifying uncertainty sources in an ensemble of hydrological climate-impact projections, Water Resources Research, 49, 1523-1536, https://doi.org/10.1029/2011WR011533, 2013.

Mboga, N., Persello, C., Bergado, J. R., and Stein, A.: Detection of informal settlements from VHR images using convolutional neural networks, Remote sensing, 9, 1106, https://doi.org/10.3390/rs9111106, 2017.

Song, T., Ding, W., Liu, H., Wu, J., Zhou, H., and Chu, J.: Uncertainty quantification in machine learning modeling for multi-step time series forecasting: Example of recurrent neural networks in discharge simulations, Water, 12, 912, https://doi.org/10.3390/w12030912, 2020.

Taylor, R., Ojha, V., Martino, I., and Nicosia, G.: Sensitivity analysis for deep learning: ranking hyper-parameter influence, 2021 IEEE 33rd International Conference on Tools with Artificial Intelligence (ICTAI), 512-516, https://doi.org/10.1109/ICTAI52525.2021.00083, 2021.

Zhang, Y., and Wallace, B.: A sensitivity analysis of (and practitioners' guide to) convolutional neural networks for sentence classification, arXiv [preprint], arXiv:1510.03820, 2015.

*For above reasons, I do not support its publication in the ESSD, without advanced approach or comprehensive evaluations of dataset at finer scale. Anyway, I still look forward to seeing the reviewers' comments and editor's decision.*

[Figure]

Figure 1. Time-series comparisons of δ¹⁸Oₚ among the built isoscape, iGCM simulations, and in-situ observations at selected stations in each sub-region.

[Figure]

Figure 1. Time-series comparisons of $\delta^{18}O_p$ among the built isoscape, iGCM simulations, and in-situ observations at selected stations in each sub-region.

[Figure]

Figure 1. Time-series comparisons of $\delta^{18}O_p$ among the built isoscape, iGCM simulations, and in-situ observations at selected stations in each sub-region.

Table 2. Correlation coefficient (CC) metrics of $\delta^{18}O_p$ series between observations and iGCM simulations and the built isoscape at all stations.

| Subregion | Station | Isoscape | CAM | GISSf | GISSn | HadAM | LMDZf | LMDZn | LMDZz | MIROC |
|-----------|---------|----------|-----|-------|-------|-------|-------|-------|-------|-------|
| NE | Changchun | 0.93 | 0.80 | 0.74 | 0.77 | 0.64 | 0.68 | 0.86 | 0.63 | 0.68 |
| NE | Haerbin | 0.89 | 0.38 | 0.33 | 0.66 | 0.36 | 0.45 | 0.53 | 0.47 | 0.46 |
| NE | Qiqihar | 0.97 | 0.71 | 0.63 | 0.78 | 0.58 | 0.74 | 0.81 | 0.79 | 0.68 |
| NE | Changbaishan | 0.69 | / | 0.46 | 0.60 | / | 0.59 | 0.72 | 0.51 | 0.58 |
| NE | Sanjiang | 0.97 | / | 0.46 | 0.51 | / | 0.55 | 0.63 | 0.65 | 0.38 |
| NE | Hailun | 0.93 | / | 0.74 | 0.73 | / | 0.72 | 0.76 | 0.74 | 0.69 |
| NE | Shenyang | 0.28 | / | 0.02 | 0.13 | / | -0.25 | 0.18 | -0.09 | 0.05 |
| NC | Jinzhou | 0.82 | -0.19 | 0.03 | 0.34 | 0.30 | 0.42 | 0.48 | -0.46 | 0.25 |
| NC | Shijiazhuang | 0.95 | 0.33 | 0.28 | 0.33 | 0.22 | 0.31 | 0.55 | 0.27 | 0.32 |
| NC | Taiyuan | 0.98 | 0.17 | 0.27 | -0.16 | -0.38 | -0.12 | -0.09 | -0.14 | 0.27 |
| NC | Tianjin | 0.95 | 0.43 | 0.51 | 0.29 | 0.41 | 0.43 | 0.66 | 0.12 | 0.40 |
| NC | Xian | 0.92 | 0.12 | -0.02 | 0.41 | -0.23 | 0.20 | 0.59 | 0.30 | 0.41 |
| NC | Yantai | 0.95 | 0.18 | 0.07 | 0.46 | 0.23 | 0.16 | 0.48 | -0.27 | 0.23 |
| NC | Zhengzhou | 0.91 | 0.07 | 0.18 | 0.49 | 0.00 | 0.22 | 0.56 | 0.08 | 0.39 |
| NC | Beijing | 0.68 | / | 0.75 | 0.23 | / | 0.78 | 0.74 | 0.60 | 0.80 |
| NC | Fengqiu | 0.90 | / | 0.05 | 0.77 | / | 0.26 | 0.66 | 0.09 | 0.20 |
| NC | Yucheng | 0.72 | / | 0.28 | 0.30 | / | 0.25 | 0.54 | 0.08 | 0.47 |
| NC | Changwu | 0.90 | / | -0.36 | 0.50 | / | -0.02 | 0.69 | 0.10 | 0.07 |
| SC | Changsha | 0.91 | 0.45 | 0.50 | 0.69 | 0.16 | 0.56 | 0.76 | 0.62 | 0.53 |
| SC | Chengdu | 0.87 | 0.43 | 0.17 | 0.67 | -0.05 | 0.39 | 0.68 | 0.66 | 0.52 |
| SC | Guilin | 0.94 | 0.59 | 0.55 | 0.86 | 0.43 | 0.71 | 0.79 | 0.79 | 0.71 |
| SC | Guiyang | 0.95 | 0.31 | 0.46 | 0.75 | 0.25 | 0.54 | 0.78 | 0.82 | 0.55 |
| SC | Kunming | 0.97 | 0.55 | 0.62 | 0.82 | 0.47 | 0.65 | 0.86 | 0.66 | 0.68 |
| SC | Liuzhou | 0.87 | 0.53 | 0.52 | 0.74 | 0.49 | 0.64 | 0.74 | 0.72 | 0.71 |
| SC | Wuhan | 0.92 | 0.37 | 0.25 | 0.64 | 0.09 | 0.36 | 0.70 | 0.64 | 0.51 |
| SC | Zunyi | 0.91 | 0.29 | 0.41 | 0.76 | 0.24 | 0.54 | 0.80 | 0.79 | 0.51 |
| SC | Yingtan | 0.63 | / | 0.07 | 0.35 | / | 0.31 | 0.22 | 0.42 | 0.18 |
| SC | Taoyuan | 0.83 | / | 0.53 | 0.82 | / | 0.56 | 0.76 | 0.78 | 0.40 |
| SC | Qianyanzhou | 0.87 | / | 0.12 | 0.59 | / | 0.40 | 0.63 | 0.59 | 0.54 |
| SC | Huitong | 0.81 | / | 0.45 | 0.77 | / | 0.69 | 0.80 | 0.78 | 0.45 |
| SC | Yanting | 0.71 | / | 0.32 | 0.43 | / | / | / | 0.72 | / |
| SC | Heshang | 0.77 | / | / | / | / | / | / | 0.73 | / |
| SC | Changsha | 0.82 | / | / | / | / | / | / | 0.79 | / |
| SE | Fuzhou | 0.89 | 0.24 | 0.09 | 0.50 | 0.24 | 0.23 | 0.37 | 0.52 | 0.33 |
| SE | Guangzhou | 0.90 | 0.36 | 0.60 | 0.60 | 0.47 | 0.61 | 0.58 | 0.44 | 0.44 |
| SE | Hong Kong | 0.80 | 0.61 | 0.60 | 0.79 | 0.64 | 0.68 | 0.78 | 0.64 | 0.62 |
| SE | Nanjing | 0.96 | 0.25 | 0.37 | 0.62 | 0.31 | 0.34 | 0.73 | 0.35 | 0.24 |
| SE | Changshu | 0.91 | / | 0.29 | 0.79 | / | 0.41 | 0.74 | 0.73 | 0.13 |
| SE | Dinghushan | 0.45 | / | 0.39 | 0.69 | / | 0.97 | 1.00 | 0.49 | 1.00 |
| SE | Ailaoshan | 0.97 | / | 0.66 | 0.87 | / | 0.70 | 0.83 | 0.14 | 0.80 |
| SE | Guangzhou | 0.74 | / | 0.47 | 0.84 | / | 0.91 | 0.83 | 0.64 | 0.77 |
| SE | Yongan | 0.43 | / | / | / | / | / | / | 0.35 | / |
| SE | Xishuangbanna | 0.96 | 0.53 | 0.82 | 0.85 | / | 0.57 | 0.78 | 0.68 | 0.78 |
| SE | Guangzhou | 0.77 | / | 0.51 | 0.83 | / | 0.79 | 0.76 | 0.73 | 0.64 |
| TP | Lhasa | 0.94 | 0.19 | 0.39 | 0.59 | 0.37 | 0.13 | 0.13 | 0.19 | 0.20 |

| Subregion | Station | Isoscape | CAM | GISSf | GISSn | HadAM | LMDZf | LMDZn | LMDZz | MIROC |
|---|---|---|---|---|---|---|---|---|---|---|
| TP | Lhasa | 0.78 | / | 0.58 | 0.59 | / | 0.36 | 0.19 | 0.27 | 0.45 |
| TP | Haibei | 0.87 | / | 0.24 | 0.30 | / | 0.64 | 0.50 | 0.43 | 0.40 |
| TP | Maoxian | 0.58 | / | -0.39 | 0.48 | / | 0.07 | 0.66 | 0.45 | 0.13 |
| TP | Gonggashan | 0.71 | / | 0.10 | 0.68 | / | 0.32 | 0.62 | 0.69 | 0.36 |
| TP | Delingha | 0.98 | 0.27 | 0.45 | 0.75 | 0.75 | 0.80 | 0.85 | 0.80 | 0.73 |
| TP | Nagqu | 0.92 | 0.17 | 0.16 | 0.64 | 0.15 | 0.15 | 0.06 | 0.46 | 0.40 |
| TP | Yushu | 0.81 | 0.00 | -0.12 | 0.37 | -0.11 | 0.18 | 0.28 | 0.46 | 0.29 |
| TP | Gaize | 0.96 | 0.34 | 0.03 | 0.44 | -0.06 | 0.36 | 0.37 | 0.43 | 0.18 |
| TP | Shiquanhe | 0.79 | -0.33 | 0.25 | 0.35 | -0.06 | 0.46 | 0.35 | 0.43 | 0.04 |
| TP | Lhasa | 0.94 | 0.37 | 0.63 | 0.74 | 0.45 | 0.45 | 0.40 | 0.42 | 0.56 |
| TP | Dingri | 0.89 | -0.08 | 0.61 | 0.18 | 0.71 | -0.33 | 0.44 | 0.37 | 0.35 |
| TP | Nyalam | 0.92 | 0.40 | 0.46 | 0.69 | -0.06 | 0.48 | 0.32 | 0.35 | 0.48 |
| TP | Tuotuohe | 0.94 | 0.18 | 0.10 | 0.64 | 0.44 | 0.66 | 0.67 | 0.67 | 0.48 |
| TP | Baidi | 0.84 | / | 0.51 | 0.62 | / | 0.29 | 0.47 | 0.44 | 0.49 |
| TP | Dui | 0.87 | / | 0.56 | 0.53 | / | 0.34 | 0.28 | 0.39 | 0.43 |
| TP | Taxkorgen | 0.97 | 0.19 | 0.70 | 0.79 | / | 0.80 | 0.82 | 0.83 | 0.70 |
| TP | Wengguo | 0.87 | / | 0.66 | 0.57 | / | 0.29 | 0.53 | 0.63 | 0.56 |
| TP | Lulang | 0.80 | / | 0.45 | 0.76 | / | 0.63 | 0.62 | 0.68 | 0.27 |
| TP | Nuxia | 0.59 | / | -0.21 | 0.79 | / | / | / | 0.49 | / |
| TP | Yeniugou | 0.84 | / | 0.88 | 0.83 | / | / | / | 0.88 | / |
| NW | Baotou | 0.93 | 0.43 | 0.52 | 0.51 | 0.39 | 0.43 | 0.60 | 0.28 | 0.46 |
| NW | Hetian | 0.96 | 0.49 | 0.52 | 0.82 | 0.55 | 0.81 | 0.89 | 0.88 | 0.82 |
| NW | Lanzhou | 0.94 | 0.18 | 0.22 | -0.07 | 0.53 | 0.60 | 0.60 | 0.45 | 0.61 |
| NW | Wulumuqi | 0.97 | 0.40 | 0.67 | 0.76 | 0.76 | 0.78 | 0.86 | 0.87 | 0.73 |
| NW | Yinchuan | 0.95 | 0.43 | 0.58 | 0.34 | 0.45 | 0.62 | 0.76 | 0.75 | 0.63 |
| NW | Zhangye | 0.96 | 0.35 | 0.59 | 0.57 | 0.62 | 0.80 | 0.83 | 0.79 | 0.76 |
| NW | Fukang | 0.94 | / | 0.75 | 0.78 | / | 0.87 | 0.88 | 0.86 | 0.84 |
| NW | Cele | 0.90 | / | -0.38 | 0.66 | / | 0.59 | 0.79 | 0.72 | 0.55 |
| NW | Linze | 0.93 | / | 0.68 | 0.67 | / | 0.78 | 0.79 | 0.68 | 0.81 |
| NW | Shapotou | 0.48 | / | 0.14 | 0.20 | / | 0.30 | 0.30 | -0.10 | 0.17 |
| NW | Ansai | 0.73 | / | -0.06 | 0.53 | / | -0.04 | 0.66 | 0.00 | 0.33 |
| NW | Erdos | 0.19 | / | 0.01 | 0.70 | / | 0.33 | 0.13 | 0.17 | -0.54 |
| NW | Naiman | 0.88 | / | 0.62 | 0.64 | / | 0.55 | 0.42 | 0.75 | 0.11 |

Table 3. Root mean square error (RMSE, ‰) metrics of $\delta^{18}O_p$ series between observations and iGCM simulations and the built isoscape at all stations.

| Subregion | Station | Isoscape | CAM | GISSf | GISSn | HadAM | LMDZf | LMDZn | LMDZz | MIROC |
|---|---|---|---|---|---|---|---|---|---|---|
| NE | Changchun | 7.47 | 11.50 | 14.71 | 14.98 | 17.35 | 16.14 | 11.45 | 22.16 | 13.80 |
| NE | Haerbin | 8.31 | 20.78 | 24.55 | 31.03 | 22.20 | 20.36 | 16.32 | 23.95 | 20.69 |
| NE | Qiqihar | 13.67 | 38.19 | 43.10 | 34.59 | 50.34 | 39.30 | 33.69 | 38.38 | 38.57 |
| NE | Changbaishan | 31.87 | / | 47.09 | 46.38 | / | 22.75 | 18.46 | 45.71 | 21.93 |
| NE | Sanjiang | 9.31 | / | 34.13 | 33.77 | / | 31.78 | 24.98 | 24.61 | 29.68 |
| NE | Hailun | 16.44 | / | 26.52 | 32.42 | / | 27.26 | 26.73 | 26.57 | 26.28 |
| NE | Shenyang | 20.70 | / | 24.61 | 20.10 | / | 22.11 | 12.67 | 33.25 | 17.39 |
| NC | Jinzhou | 5.46 | 10.88 | 8.25 | 6.59 | 7.97 | 6.56 | 6.68 | 21.52 | 11.30 |
| NC | Shijiazhuang | 15.12 | 56.29 | 49.40 | 45.68 | 40.41 | 44.44 | 36.17 | 83.95 | 52.45 |
| NC | Taiyuan | 5.37 | 22.94 | 23.43 | 24.56 | 20.93 | 19.33 | 20.78 | 33.76 | 19.09 |
| NC | Tianjin | 9.34 | 31.11 | 25.51 | 24.90 | 25.79 | 24.63 | 18.78 | 54.64 | 31.35 |
| NC | Xian | 10.27 | 40.99 | 35.42 | 32.86 | 32.12 | 28.35 | 20.92 | 30.46 | 26.61 |
| NC | Yantai | 5.84 | 19.21 | 21.43 | 18.19 | 18.76 | 20.83 | 17.23 | 36.96 | 24.88 |
| NC | Zhengzhou | 10.99 | 36.21 | 32.11 | 23.59 | 26.80 | 25.12 | 20.52 | 42.04 | 23.46 |
| NC | Beijing | 28.37 | / | 20.28 | 27.08 | / | 14.61 | 14.19 | 29.14 | 10.71 |
| NC | Fengqiu | 10.03 | / | 22.74 | 17.20 | / | 18.24 | 14.66 | 27.88 | 19.44 |
| NC | Yucheng | 13.78 | / | 21.10 | 19.83 | / | 17.29 | 15.57 | 31.43 | 15.38 |
| NC | Changwu | 13.79 | / | 40.64 | 37.52 | / | 21.68 | 13.69 | 31.02 | 28.30 |
| SC | Changsha | 9.74 | 21.83 | 23.99 | 22.08 | 27.34 | 19.65 | 15.59 | 19.36 | 19.72 |
| SC | Chengdu | 16.55 | 56.26 | 56.99 | 62.06 | 39.34 | 29.57 | 25.76 | 26.21 | 43.54 |
| SC | Guilin | 10.18 | 24.79 | 24.69 | 19.40 | 29.09 | 21.52 | 19.23 | 18.46 | 21.60 |
| SC | Guiyang | 9.14 | 31.95 | 26.87 | 23.28 | 30.90 | 25.78 | 19.23 | 22.24 | 25.05 |
| SC | Kunming | 16.06 | 47.73 | 44.50 | 44.59 | 44.75 | 47.67 | 40.91 | 47.68 | 45.03 |
| SC | Liuzhou | 11.06 | 20.81 | 19.86 | 16.67 | 21.12 | 18.23 | 16.09 | 17.71 | 16.44 |
| SC | Wuhan | 7.79 | 19.69 | 24.96 | 19.37 | 24.38 | 20.34 | 15.64 | 16.61 | 17.62 |
| SC | Zunyi | 13.82 | 38.47 | 33.45 | 31.05 | 34.47 | 28.45 | 21.06 | 22.14 | 28.87 |
| SC | Yingtan | 17.72 | / | 22.76 | 25.17 | / | 13.90 | 19.02 | 22.05 | 14.57 |
| SC | Taoyuan | 11.19 | / | 17.54 | 15.16 | / | 13.78 | 11.46 | 14.23 | 16.38 |
| SC | Qianyanzhou | 7.51 | / | 20.86 | 18.09 | / | 14.37 | 13.41 | 13.93 | 13.33 |
| SC | Huitong | 12.06 | / | 18.17 | 15.19 | / | 10.52 | 9.30 | 14.60 | 15.35 |
| SC | Yanting | 21.13 | / | 31.54 | 33.49 | / | / | / | 20.09 | / |
| SC | Heshang | 20.75 | / | / | / | / | / | / | 21.97 | / |
| SC | Changsha | 19.15 | / | / | / | / | / | / | 21.62 | / |
| SE | Fuzhou | 10.88 | 27.18 | 28.32 | 22.42 | 27.09 | 28.15 | 25.79 | 21.75 | 25.71 |
| SE | Guangzhou | 6.39 | 15.12 | 10.76 | 10.91 | 16.41 | 11.04 | 13.11 | 17.21 | 13.25 |
| SE | Hong Kong | 38.15 | 43.96 | 47.98 | 34.52 | 50.52 | 42.01 | 32.63 | 45.53 | 40.70 |
| SE | Nanjing | 6.91 | 22.14 | 19.21 | 17.01 | 23.01 | 25.58 | 20.30 | 22.12 | 24.79 |
| SE | Changshu | 3.50 | / | 9.88 | 5.73 | / | 14.49 | 10.67 | 6.84 | 11.59 |
| SE | Dinghushan | 22.85 | / | 15.51 | 14.40 | / | 3.71 | 1.20 | 22.44 | 5.39 |
| SE | Ailaoshan | 6.46 | / | 20.21 | 15.92 | / | 27.27 | 24.72 | 33.50 | 25.93 |
| SE | Guangzhou | 20.37 | / | 16.56 | 9.74 | / | 5.71 | 6.20 | 23.52 | 6.89 |
| SE | Yongan | 15.83 | / | / | / | / | / | / | 16.42 | / |
| SE | Xishuangbanna | 9.26 | 20.69 | 19.55 | 16.55 | / | 32.37 | 29.43 | 28.11 | 32.21 |
| SE | Guangzhou | 11.43 | / | 16.95 | 11.17 | / | 9.01 | 8.77 | 12.06 | 9.43 |
| TP | Lhasa | 16.48 | 45.39 | 48.24 | 40.56 | 42.23 | 50.30 | 53.37 | 43.84 | 46.20 |

| Subregion | Station | Isoscape | CAM | GISSf | GISSn | HadAM | LMDZf | LMDZn | LMDZz | MIROC |
|---|---|---|---|---|---|---|---|---|---|---|
| TP | Lhasa | 24.27 | / | 30.61 | 30.51 | / | 34.27 | 37.35 | 36.78 | 30.10 |
| TP | Haibei | 11.23 | / | 38.38 | 45.39 | / | 21.11 | 19.96 | 33.95 | 29.98 |
| TP | Maoxian | 13.61 | / | 25.84 | 29.51 | / | 13.38 | 9.50 | 15.67 | 12.89 |
| TP | Gonggashan | 27.37 | / | 26.75 | 31.77 | / | 26.49 | 24.21 | 30.24 | 23.97 |
| TP | Delingha | 14.94 | 87.68 | 127.83 | 113.21 | 53.20 | 40.92 | 35.78 | 85.71 | 71.41 |
| TP | Nagqu | 16.24 | 40.61 | 42.90 | 35.18 | 25.44 | 61.90 | 62.31 | 40.53 | 35.25 |
| TP | Yushu | 11.73 | 32.84 | 50.28 | 37.89 | 13.08 | 31.88 | 29.51 | 20.69 | 29.43 |
| TP | Gaize | 13.95 | 44.70 | 72.06 | 53.95 | 31.61 | 51.14 | 50.41 | 40.32 | 47.10 |
| TP | Shiquanhe | 20.98 | 37.84 | 33.23 | 32.83 | 30.81 | 50.83 | 49.15 | 35.91 | 51.55 |
| TP | Lhasa | 22.86 | 48.17 | 51.15 | 47.12 | 41.72 | 75.87 | 79.16 | 57.43 | 61.10 |
| TP | Dingri | 9.82 | 20.02 | 17.23 | 25.29 | 13.54 | 41.22 | 40.39 | 20.09 | 39.40 |
| TP | Nyalam | 22.22 | 44.29 | 48.22 | 37.36 | 65.87 | 56.07 | 58.01 | 52.53 | 64.70 |
| TP | Tuotuohe | 22.46 | 84.32 | 112.41 | 78.12 | 48.50 | 56.33 | 55.78 | 45.39 | 60.21 |
| TP | Baidi | 12.24 | / | 21.66 | 20.40 | / | 33.47 | 32.83 | 23.63 | 36.25 |
| TP | Dui | 17.57 | / | 28.75 | 28.00 | / | 43.04 | 43.72 | 34.06 | 44.76 |
| TP | Taxkorgen | 12.64 | 13.27 | 44.91 | 48.34 | / | 31.72 | 30.37 | 32.65 | 34.84 |
| TP | Wengguo | 10.97 | / | 15.37 | 17.58 | / | 29.67 | 29.66 | 19.67 | 31.43 |
| TP | Lulang | 39.66 | / | 29.51 | 25.03 | / | 27.43 | 26.70 | 44.52 | 22.53 |
| TP | Nuxia | 25.64 | / | 10.31 | 14.28 | / | / | / | 24.23 | / |
| TP | Yeniugou | 16.60 | / | 30.61 | 37.42 | / | / | / | 16.30 | / |
| NW | Baotou | 11.33 | 50.59 | 42.77 | 38.48 | 43.78 | 32.82 | 24.41 | 60.87 | 40.64 |
| NW | Hetian | 19.61 | 60.60 | 67.10 | 77.90 | 76.83 | 36.28 | 31.10 | 61.56 | 38.01 |
| NW | Lanzhou | 10.60 | 55.48 | 53.01 | 65.13 | 28.74 | 24.34 | 23.92 | 38.27 | 42.90 |
| NW | Wulumuqi | 17.15 | 66.80 | 59.34 | 69.90 | 63.51 | 55.43 | 44.56 | 69.41 | 49.72 |
| NW | Yinchuan | 8.14 | 38.30 | 33.31 | 35.84 | 27.23 | 21.02 | 18.36 | 33.55 | 29.89 |
| NW | Zhangye | 20.80 | 84.71 | 91.43 | 93.86 | 53.70 | 40.24 | 37.27 | 65.15 | 64.30 |
| NW | Fukang | 17.91 | / | 35.66 | 46.33 | / | 23.33 | 21.36 | 40.93 | 23.37 |
| NW | Cele | 8.46 | / | 49.74 | 52.79 | / | 18.44 | 17.65 | 32.59 | 25.31 |
| NW | Linze | 16.52 | / | 48.37 | 51.71 | / | 21.65 | 21.60 | 47.16 | 30.30 |
| NW | Shapotou | 18.98 | / | 41.20 | 52.35 | / | 14.44 | 16.28 | 41.61 | 23.84 |
| NW | Ansai | 18.09 | / | 38.55 | 32.02 | / | 24.17 | 13.92 | 37.28 | 23.82 |
| NW | Erdos | 14.76 | / | 22.65 | 20.64 | / | 7.69 | 9.35 | 24.33 | 11.91 |
| NW | Naiman | 12.76 | / | 13.85 | 13.75 | / | 12.61 | 12.42 | 36.06 | 14.49 |

[Figure]

Figure 2. Spatial distributions of δ¹⁸Oₚ in each month for the period of 1979-2007 as obtained from observations (circles), the built isoscape (left column), and better-performing iGCMs (right three columns).

[Figure]

Figure 1. Spatial distributions of $\delta^{18}O_p$ in each month for the period of 1979-2007 as obtained from observations (circles), the built isoscape (left column), and better-performing iGCMs (right three columns).

Table 4. Detailed information and sources for each observation site.

| Subregion | Site | Latitude | Longitude | Altitude (m) | Period | Data volume | Data source |
|---|---|---|---|---|---|---|---|
| NE | Changchun | 43.90 | 125.22 | 237 | 1999-2001 | 22 | GNIP |
| NE | Haerbin | 45.68 | 126.62 | 172 | 1986-1997 | 35 | GNIP |
| NE | Qiqihar | 47.38 | 123.92 | 147 | 1988-1992 | 50 | GNIP |
| NE | Changbaishan | 42.40 | 128.11 | 738.1 | 2005-2010 | 72 | CHNIP |
| NE | Sanjiang | 47.35 | 133.30 | 55 | 2005-2007 | 24 | CHNIP |
| NE | Hailun | 47.45 | 126.93 | 236 | 2005-2009 | 32 | CHNIP |
| NE | Shenyang | 41.52 | 123.37 | 49 | 2005-2010 | 33 | CHNIP |
| NC | Jinzhou | 41.13 | 121.10 | 66 | 1987-1989 | 12 | GNIP |
| NC | Shijiazhuang | 38.03 | 114.42 | 80 | 1985-2003 | 146 | GNIP |
| NC | Taiyuan | 37.78 | 112.55 | 778 | 1986-1988 | 20 | GNIP |
| NC | Tianjin | 39.10 | 117.17 | 3 | 1988-2001 | 64 | GNIP |
| NC | Xian | 34.30 | 108.93 | 397 | 1985-1992 | 60 | GNIP |
| NC | Yantai | 37.53 | 121.40 | 47 | 1986-1991 | 44 | GNIP |
| NC | Zhengzhou | 34.72 | 113.65 | 110 | 1985-1992 | 57 | GNIP |
| NC | Beijing | 39.96 | 115.43 | 1248 | 2005-2010 | 44 | CHNIP |
| NC | Fengqiu | 35.01 | 114.33 | 67.5 | 2005-2007 | 22 | CHNIP |
| NC | Yucheng | 36.83 | 116.57 | 22 | 2005-2008 | 27 | CHNIP |
| NC | Changwu | 35.24 | 107.68 | 1200 | 2005-2009 | 32 | CHNIP |
| SC | Changsha | 28.20 | 113.07 | 37 | 1988-1992 | 57 | GNIP |
| SC | Chengdu | 30.67 | 104.02 | 506 | 1986-1998 | 67 | GNIP |
| SC | Guilin | 25.07 | 110.08 | 170 | 1983-1990 | 92 | GNIP |
| SC | Guiyang | 26.58 | 106.72 | 1071 | 1988-1992 | 58 | GNIP |
| SC | Kunming | 25.02 | 102.68 | 1892 | 1986-2003 | 152 | GNIP |
| SC | Liuzhou | 24.35 | 109.40 | 97 | 1988-1992 | 53 | GNIP |
| SC | Wuhan | 30.62 | 114.13 | 23 | 1986-1998 | 50 | GNIP |
| SC | Zunyi | 27.70 | 106.88 | 844 | 1986-1992 | 74 | GNIP |
| SC | Yingtan | 28.12 | 116.56 | 45 | 2005-2010 | 56 | CHNIP |
| SC | Taoyuan | 28.93 | 111.44 | 106 | 2005-2010 | 47 | CHNIP |
| SC | Qianyanzhou | 26.44 | 115.03 | 76.4 | 2005-2007 | 33 | CHNIP |
| SC | Huitong | 26.85 | 109.61 | 541 | 2005-2010 | 48 | CHNIP |
| SC | Yanting | 31.27 | 105.46 | 420 | 2009-2010 | 18 | CHNIP |
| SC | Heshang | 30.45 | 110.42 | 270 | 2011-2018 | 80 | Wang et al. (2020) |
| SC | Changsha | 28.25 | 112.55 | 37 | 2010-2017 | 95 | Zhang (2019) |
| SE | Fuzhou | 26.08 | 119.28 | 16 | 1985-1992 | 71 | GNIP |
| SE | Guangzhou | 23.13 | 113.32 | 7 | 1986-1989 | 30 | GNIP |
| SE | Hong Kong | 22.32 | 114.17 | 66 | 1961-2018 | 549 | GNIP |
| SE | Nanjing | 32.18 | 118.18 | 26 | 1987-1992 | 58 | GNIP |
| SE | Changshu | 31.33 | 120.42 | 3.1 | 2005-2006 | 21 | CHNIP |
| SE | Dinghushan | 23.16 | 112.55 | 90 | 2005-2010 | 24 | CHNIP |
| SE | Ailaoshan | 24.55 | 101.03 | 2481 | 2005-2007 | 28 | CHNIP |
| SE | Guangzhou | 23.13 | 113.32 | 7 | 2007-2014 | 94 | Yang |
| SE | Yongan | 28.89 | 120.85 | 23 | 2014-2018 | 34 | Chen (2020) |

| Subregion | Site | Latitude | Longitude | Altitude (m) | Period | Data volume | Data source |
|---|---|---|---|---|---|---|---|
| SE | Xishuangbanna | 21.93 | 101.27 | 750 | 2002-2004 | 33 | Liu et al. (2007) |
| SE | Guangzhou | 23.15 | 113.35 | 39 | 2007-2009 | 34 | Xie et al. (2011) |
| TP | Lhasa | 29.70 | 91.13 | 3649 | 1986-1992 | 42 | GNIP |
| TP | Lhasa | 29.41 | 91.21 | 3688 | 2005-2009 | 32 | CHNIP |
| TP | Haibei | 37.56 | 101.31 | 3280 | 2005-2009 | 23 | CHNIP |
| TP | Maoxian | 31.70 | 103.90 | 1826 | 2005-2009 | 22 | CHNIP |
| TP | Gonggashan | 29.58 | 102.00 | 2950 | 2005-2010 | 52 | CHNIP |
| TP | Delingha | 37.37 | 97.37 | 2981 | 1992-2006 | 115 | TNIP |
| TP | Nagqu | 31.48 | 92.07 | 4508 | 1999-2005 | 59 | TNIP |
| TP | Yushu | 33.02 | 97.02 | 3682 | 2000-2004 | 37 | TNIP |
| TP | Gaize | 32.30 | 84.07 | 4430 | 1998-2005 | 45 | TNIP |
| TP | Shiquanhe | 32.50 | 80.08 | 4278 | 1999-2002 | 24 | TNIP |
| TP | Lhasa | 29.70 | 91.13 | 3658 | 1994-2006 | 85 | TNIP |
| TP | Dingri | 28.65 | 87.12 | 4330 | 2000-2006 | 21 | TNIP |
| TP | Nyalam | 28.18 | 85.97 | 3810 | 1996-2006 | 78 | TNIP |
| TP | Tuotuohe | 34.22 | 92.43 | 4533 | 1991-2005 | 104 | TNIP |
| TP | Baidi | 29.12 | 90.43 | 4430 | 2004-2007 | 22 | TNIP |
| TP | Dui | 28.58 | 90.53 | 5030 | 2004-2007 | 24 | TNIP |
| TP | Taxkorgen | 37.77 | 75.27 | 3100 | 2003-2005 | 22 | TNIP |
| TP | Wengguo | 28.90 | 90.35 | 4500 | 2004-2007 | 14 | TNIP |
| TP | Lulang | 29.77 | 94.73 | 3330 | 2007-2014 | 79 | Yang |
| TP | Nuxia | 29.47 | 94.65 | 2920 | 2009-2014 | 43 | Yang |
| TP | Yeniugou | 38.46 | 99.54 | 3320 | 2008-2009 | 13 | Zhao et al. (2011) |
| NW | Baotou | 40.67 | 109.85 | 1067 | 1986-1992 | 61 | GNIP |
| NW | Hetian | 37.13 | 79.93 | 1375 | 1988-1992 | 47 | GNIP |
| NW | Lanzhou | 36.05 | 103.88 | 1517 | 1985-1999 | 41 | GNIP |
| NW | Wulumuqi | 43.78 | 87.62 | 918 | 1986-2003 | 131 | GNIP |
| NW | Yinchuan | 38.48 | 106.22 | 1112 | 1988-2000 | 30 | GNIP |
| NW | Zhangye | 38.93 | 100.43 | 1483 | 1986-2003 | 86 | GNIP |
| NW | Fukang | 44.29 | 87.93 | 460 | 2005-2009 | 47 | CHNIP |
| NW | Cele | 37.02 | 80.73 | 1306 | 2005-2007 | 13 | CHNIP |
| NW | Linze | 39.35 | 100.13 | 1375 | 2005-2009 | 28 | CHNIP |
| NW | Shapotou | 37.28 | 105.00 | 1350 | 2005-2010 | 28 | CHNIP |
| NW | Ansai | 36.86 | 109.32 | 1083 | 2005-2010 | 46 | CHNIP |
| NW | Erdos | 39.49 | 110.19 | 1270 | 2006-2009 | 11 | CHNIP |
| NW | Naiman | 42.93 | 120.70 | 363 | 2005-2010 | 28 | CHNIP |

---

## Author Comment (AC3)

**Responses to CC2:**

*General comments:*
*Obtaining a high-resolution long-term precipitation oxygen isoscape dataset can be critical for relevant hydrological studies. This study presented a first attempt to solve this issue, really appreciate, but from my point of view, it is still a rather premature dataset and have limited value. The authors downscaled and fused eight iGCMs precipitation oxygen isoscape using five different methods from a coarse spatial resolution (~2/3 degree) to a higher spatial resolution (0.5 degree). However, the work is not innovative and no important and robust findings were obtained.*

**Re:** We would like to thank the reviewer for the time taking in reviewing our manuscript and providing constructive comments. The crux of the paper is to develop a long-term high-spatiotemporal resolution dataset for the mainland of China using an optimal hybrid approach, which is the best of the existing datasets available so far in the country. Following these suggestions, we have carefully revised the manuscript, especially for the validation of the dataset's quality and presenting the robust findings. Below please find our point-by-point responses to these comments.

   With the motivations of resolving the lack and uneven distribution of observations, as well as the coarse and biased iGCM simulations, this study takes a hybrid approach that makes full use of observations to integrate the advantages of various iGCMs by using the optimal combination of data fusion and bias correction methods. In other words, the devised approach used all observations and iGCM simulations to the utmost extent to ensure the highest accuracy possible throughout the entire time period. This is a first attempt to develop such a dataset in China, which is of great importance in providing a data foundation for the study of complex hydrological and climatic systems.

   The fusion and bias correction methods in the study have been widely used; however, to the best of our knowledge they have not been used in the field of stable isotopes. This study first compared the performance of these methods to find the best combination to build the dataset for different time periods by regions. The results showed that the CNN fusion method consistently performed the best for all sub-regions in China, and two bias correction methods (LS and DT) showed similar performance. The combination of the CNN fusion and bias correction methods is satisfactory to develop a high-resolution isoscape with a long time period. In other words, in order to maximize the utilization of available data, the CNN fusion method was used for the common period of all climate simulation and observations, while the bias correction methods were used for the periods with only one or two climate simulations, and with no observations. Considering China is a large country with various climatic conditions and complex terrain, the selected methods may also be able to be applied to other regions of the world.

   Definitely, we agree with the reviewer that the dataset quality is the primary concern, and the quality was not well evaluated in the original manuscript. In the revision of this manuscript, following the reviewers' specific comments, a comprehensive assessment of the dataset at a finer scale was conducted. Specifically, the dataset was evaluated with respect to reproducing the time-series of in-situ observations at station scale, as well as the spatial pattern for each month. The results show that the in-situ observations were reasonably represented by the isoscape time series for all stations. The correlation coefficients between the isoscape simulations and observations are larger than 0.8 for 73% of the stations and larger than 0.9 for 49% of the stations. The root mean square errors between the isoscape simulations and observations are smaller than 20‰ for 78% of the stations and less than 15‰ for 56% of the stations. From the monthly spatial distribution of isotopes, CNN fusion simulations also capture the spatial pattern of observations most accurately. All results

showed that the isoscape dataset is of high quality. **The detailed results can be found in the responses to specific comments below.**

*The results do not convince me since this method highly depends on the training data, which cover a short period, are unevenly distributed across regions, and insufficient to train model.*

**Re:** We also agree with the reviewer that the time period of observed data is short for some stations. However, the monthly data were used in our study. More importantly, to partly solve the problem of short period, the data fusion and bias correction were conducted at the regional scale for a specific season to pool multiple sub-datasets together for training the neural network models and bias correction methods. In other words, the training of model was not conducted for each individual station. Moreover, considering the lack of observed data for training, we chose the simple structure to make the network not too deep with desirable results. For example, we chose the 1D convolutional neural network because of its advantages for data scarcity (Kiranyaz et al., 2021). Moreover, we only used the convolutional neural networks (CNN) fusion method to generate the dataset for the 1969-2007 period, which is more dependent on training data. For this period, the observed and iGCM simulations are relatively abundant. In other periods, bias correction methods were used, which are relatively less dependent on observations. Moreover, from the refined dataset evaluation (detailed results can be seen in the last part), the in-situ observations were reasonably represented by the isoscape time series at station scale, and the spatial pattern was captured most accurately for each month. All results showed that the CNN fusion method performs well, and the established dataset is of high quality and reliable.

Definitely, the models might perform even better if more observations were available. But then again, the lack of observations is the main motivation to develop this dataset, as we were attempted to maximize the utilization of available data from all observations and climate model simulations. In addition, we will introduce some physical-based ancillary data in the fusion processes, such as elevation and meteorological data, to fine tune the data and make our dataset more reliable.

References:

Kiranyaz, S., Avci, O., Abdeljaber, O., Ince, T., Gabbouj, M., and Inman, D. J.: 1D convolutional neural networks and applications: A survey, Mechanical systems and signal processing, 151, 107398, https://doi.org/10.1016/j.ymssp.2020.107398, 2021.

*Moreover, as we know, the precipitation oxygen isoscape is highly dependent on local climate conditions, terrain factors, as well as large-scale atmospheric and local circulation. No such physical-based ancillary data were used in this study, which limited the further applications of produced 0.5-degree data. The sub-pixel spatial patterns within a coarse pixel changes, but the current methodology cannot get this information.*

**Re:** Thank you for your suggestions. We agree with the review that the local climatic conditions, terrain and other factors may need to be considered in data fusion. In the revised manuscript, we will introduce some physical-based ancillary data in the fusion, such as elevation and meteorological data, to enrich the information in the process of data fusion. These data are closely related to precipitation isotopes and are easily accessible.

The considered ancillary data include Shuttle Radar Topography Mission digital elevation data (SRTM DEM) with the spatial resolution of 90m, and dataset of gridded monthly precipitation and temperature in China with spatial resolution of the 0.5 degree.

The generation of isoscape taking into account ancillary data can be divided into five steps. (1) Prior to generating the dataset, inverse distance weighting (IDW) method was used to interpolate all iGCM simulations and ancillary data to observation stations. (2) Three neural network data fusion and two bias correction methods were trained using observations and iGCM simulations for all months within a season and all stations within a sub-region. For the fusion methods, ancillary data were also included in the training process. (3) The performance of each model was evaluated for the validation period by the cross-validation method to find the optimal data fusion and bias correction methods. (4) All iGCM simulations and ancillary data were interpolated to the LMDZ4 zoomed grid with a spatial resolution of approximately 50 km by the IDW method. (5) The optimal trained model and bias correction methods were applied to all grid points within a region and all months within a season.

*For the implementation of models, uncertainty of the datasets resulting from the model structures, parameters, training and testing strategies is not even discussed.*

**Re:** In the original manuscript, the uncertainty was partly analyzed, but not comprehensive. The plotted +/- one standard deviation in Figs. 5-6 shows the dispersion degree of the evaluation metrics (CC and RMSE) for the simulated results of all bias correction and data fusion methods over 100 trials. The standard deviations of CC and RMSE calculated by LS and DT corrected simulations are smaller, while those calculated by BP, LSTM and CNN fused simulations are larger. The standard deviation of CC and RMSE calculated by the CNN fused simulations is the smallest among fusion methods. It can be considered that LS and DT correction methods show smaller uncertainties in CC and RMSE than the BP, LSTM and CNN fusion methods. CNN fusion methods show smaller uncertainties than the other two fusion methods.

In the revised manuscript, an uncertainty analysis for the use of data fusion methods was further analyzed. Specifically, the CNN method was taken as an example to analyze the uncertainty derived from model structure, model parameters and training samples for fusing isotope in South China (SC) over summer (JJA). For the model structure, a different number of convolution layers were selected, because the model is very sensitive to the number of convolution layers (Mboga et al., 2017). For model parameters, three parameters, namely learning rate, batch size and filter size, were selected based on previous sensitivity studies (Zhang and Wallace, 2015; Taylor et al., 2021; Mboga et al., 2017; Bengio, 2012). Commonly-used values for each parameter were selected to form twelve groups of parameter setting schemes. For the training sample, five different training-test sets were randomly generated. To sum up, the modelling combination scheme for uncertainty analysis is shown in Table 1.

A variance decomposition method (Song et al., 2020; Bosshard et al., 2013) was used to calculate the uncertainty contribution for these three sources as well as their interactions. The correlation coefficient (CC) and root mean square error (RMSE) were used as the evaluation criteria. Then, all the combination schemes were trained and the evaluation criteria were calculated, which was repeated 30 times for each combination. Results shows that the accuracy (i.e. standard error) of CC and RMSE are respectively 0.0025 and 0.0127‰. The relative contribution of each source to the total uncertainty is shown in Fig. 1. As can be seen, there is little difference in the relative contribution of uncertainty sources between the two evaluation criteria. Model parameters have the greatest contribution to the total uncertainty with the contribution being more than 50%, while the contribution of training samples is the least, which is less than 1%. These results indicate that the model is robust and not very dependent on training data.

Table 1. The modeling combination scheme of uncertainty calculation.

| Model structure | Model parameters | | | Training samples |
|---|---|---|---|---|
| Number of convolutional layer | Learning rate | Batch size | Filter size | |
| 1 | 0.0005 | 20 | 3 | |
| 2 | 0.001 | 50 | 4 | Samples 1-5 |
| 3 | 0.002 | | | |

[Figure]

Figure 1. The relative contribution of each source to the total uncertainty.

References:

Bengio, Y.: Practical recommendations for gradient-based training of deep architectures, Neural networks: Tricks of the trade, Springer, Berlin, Heidelberg, 437-478, https://doi.org/10.1007/978-3-642-35289-8_26, 2012.

Bosshard, T., Carambia, M., Goergen, K., Kotlarski, S., Krahe, P., Zappa, M., and Schär, C.: Quantifying uncertainty sources in an ensemble of hydrological climate-impact projections, Water Resources Research, 49, 1523-1536, https://doi.org/10.1029/2011WR011533, 2013.

Mboga, N., Persello, C., Bergado, J. R., and Stein, A.: Detection of informal settlements from VHR images using convolutional neural networks, Remote sensing, 9, 1106, https://doi.org/10.3390/rs9111106, 2017.

Song, T., Ding, W., Liu, H., Wu, J., Zhou, H., and Chu, J.: Uncertainty quantification in machine learning modeling for multi-step time series forecasting: Example of recurrent neural networks in discharge simulations, Water, 12, 912, https://doi.org/10.3390/w12030912, 2020.

Taylor, R., Ojha, V., Martino, I., and Nicosia, G.: Sensitivity analysis for deep learning: ranking hyper-parameter influence, 2021 IEEE 33rd International Conference on Tools with Artificial Intelligence (ICTAI), 512-516, https://doi.org/10.1109/ICTAI52525.2021.00083, 2021.

Zhang, Y., and Wallace, B.: A sensitivity analysis of (and practitioners' guide to) convolutional neural networks for sentence classification, arXiv [preprint], arXiv:1510.03820, 2015.

*Data-quality assessment at finer scale is poor presented. I have concerns about the reliability of spatial-temporal variations in your new data product at fine resolution.*

**Re:** Thanks lot for the insightful comments. **Following the reviewer's comments, a comprehensive**

**assessment of the dataset at the finer scale was conducted.** Specifically, the dataset was evaluated with respect to reproducing the time-series of in-situ observations at station scale, as well as the spatial pattern for each month. The detailed procedures are presented as follows.

To evaluate the dataset for all stations, the correlation coefficient (CC) and root mean square error (RMSE) were calculated for $\delta^{18}O_p$ series between observations and raw iGCM simulations, and between observations and built isoscape for all stations over the common period (Tables 2-3). The results show that the built isoscape performs excellent for the vast majority of stations, with larger CCs and smaller RMSEs than iGCM simulations. Specifically, the CCs between the isoscape simulations and observations are larger than 0.8 for 73% of the stations and larger than 0.9 for 49% of the stations. The RMSEs between the isoscape simulations and observations are smaller than 20‰ for 78% of the stations and less than 15‰ for 56% of the stations.

To further demonstrate the dataset quality, two stations with appropriate length of observation were randomly selected for each sub-region. Totally, 12 stations were selected. The time series of $\delta^{18}O_p$ were plotted for observations, iGCM simulations, and the generated isoscape (Fig. 2). As can be seen from Fig. 2, the variations of $\delta^{18}O_p$ are very consistent between the generated isoscape and observations, and the isoscape performs much better than raw iGCM simulations. In particular for the period before 2007, the CNN model integrates the advantages of various simulations and captures most features of the observed data. These results generally prove that the generated isoscape is reliable.

Fig. 3 further shows the monthly spatial distribution of observed, newly generated, and better-performing simulated $\delta^{18}O_p$ for their common period (i.e. 1979-2007). The spatial pattern presented by the built isoscape shows the best consistency with the observations. The strength of the CNN model has been demonstrated, which can make good use of the advantages of each simulation to accurately capture the characteristics of observations. For example, the LMDZ nudged model shows a strong ability to reproduce the spatial distribution of $\delta^{18}O_p$ for the eastern region in summer and autumn, but a slightly poor performance in the Qinghai-Tibet Plateau. The built isoscape combines LMDZ nudged with GISS nudged and LMDZ zoomed simulations, which show reasonably performance for the Qinghai-Tibet Plateau, and well reproduces the spatial distribution of $\delta^{18}O_p$ for mainland China.

In addition, based on suggestions from other reviewers, we will introduce some physical-based ancillary data in the fusion, such as elevation and meteorological data. Taking into account the effects of elevation and meteorological factors on precipitation isotopes might make our dataset more reliable.

**From above analyzes, we are very confident that the generated isoscape is of high quality and the best dataset available in China, which will be widely used in the future. All above results will be added in the revised manuscript.**

[Figure]

Figure 2. Time-series comparisons of $\delta^{18}O_p$ among the built isoscape, iGCM simulations, and in-situ observations at selected stations in each sub-region.

[Figure]

Figure 2. Time-series comparisons of $\delta^{18}O_p$ among the built isoscape, iGCM simulations, and in-situ observations at selected stations in each sub-region.

[Figure]

Figure 2. Time-series comparisons of $\delta^{18}O_p$ among the built isoscape, iGCM simulations, and in-situ observations at selected stations in each sub-region.

Table 2. Correlation coefficient (CC) metrics of $\delta^{18}O_p$ series between observations and iGCM simulations and the built isoscape at all stations.

| Subregion | Station | Isoscape | CAM | GISSf | GISSn | HadAM | LMDZf | LMDZn | LMDZz | MIROC |
|---|---|---|---|---|---|---|---|---|---|---|
| NE | Changchun | 0.93 | 0.80 | 0.74 | 0.77 | 0.64 | 0.68 | 0.86 | 0.63 | 0.68 |
| NE | Haerbin | 0.89 | 0.38 | 0.33 | 0.66 | 0.36 | 0.45 | 0.53 | 0.47 | 0.46 |
| NE | Qiqihar | 0.97 | 0.71 | 0.63 | 0.78 | 0.58 | 0.74 | 0.81 | 0.79 | 0.68 |
| NE | Changbaishan | 0.69 | / | 0.46 | 0.60 | / | 0.59 | 0.72 | 0.51 | 0.58 |
| NE | Sanjiang | 0.97 | / | 0.46 | 0.51 | / | 0.55 | 0.63 | 0.65 | 0.38 |
| NE | Hailun | 0.93 | / | 0.74 | 0.73 | / | 0.72 | 0.76 | 0.74 | 0.69 |
| NE | Shenyang | 0.28 | / | 0.02 | 0.13 | / | -0.25 | 0.18 | -0.09 | 0.05 |
| NC | Jinzhou | 0.82 | -0.19 | 0.03 | 0.34 | 0.30 | 0.42 | 0.48 | -0.46 | 0.25 |
| NC | Shijiazhuang | 0.95 | 0.33 | 0.28 | 0.33 | 0.22 | 0.31 | 0.55 | 0.27 | 0.32 |
| NC | Taiyuan | 0.98 | 0.17 | 0.27 | -0.16 | -0.38 | -0.12 | -0.09 | -0.14 | 0.27 |
| NC | Tianjin | 0.95 | 0.43 | 0.51 | 0.29 | 0.41 | 0.43 | 0.66 | 0.12 | 0.40 |
| NC | Xian | 0.92 | 0.12 | -0.02 | 0.41 | -0.23 | 0.20 | 0.59 | 0.30 | 0.41 |
| NC | Yantai | 0.95 | 0.18 | 0.07 | 0.46 | 0.23 | 0.16 | 0.48 | -0.27 | 0.23 |
| NC | Zhengzhou | 0.91 | 0.07 | 0.18 | 0.49 | 0.00 | 0.22 | 0.56 | 0.08 | 0.39 |
| NC | Beijing | 0.68 | / | 0.75 | 0.23 | / | 0.78 | 0.74 | 0.60 | 0.80 |
| NC | Fengqiu | 0.90 | / | 0.05 | 0.77 | / | 0.26 | 0.66 | 0.09 | 0.20 |
| NC | Yucheng | 0.72 | / | 0.28 | 0.30 | / | 0.25 | 0.54 | 0.08 | 0.47 |
| NC | Changwu | 0.90 | / | -0.36 | 0.50 | / | -0.02 | 0.69 | 0.10 | 0.07 |
| SC | Changsha | 0.91 | 0.45 | 0.50 | 0.69 | 0.16 | 0.56 | 0.76 | 0.62 | 0.53 |
| SC | Chengdu | 0.87 | 0.43 | 0.17 | 0.67 | -0.05 | 0.39 | 0.68 | 0.66 | 0.52 |
| SC | Guilin | 0.94 | 0.59 | 0.55 | 0.86 | 0.43 | 0.71 | 0.79 | 0.79 | 0.71 |
| SC | Guiyang | 0.95 | 0.31 | 0.46 | 0.75 | 0.25 | 0.54 | 0.78 | 0.82 | 0.55 |
| SC | Kunming | 0.97 | 0.55 | 0.62 | 0.82 | 0.47 | 0.65 | 0.86 | 0.66 | 0.68 |
| SC | Liuzhou | 0.87 | 0.53 | 0.52 | 0.74 | 0.49 | 0.64 | 0.74 | 0.72 | 0.71 |
| SC | Wuhan | 0.92 | 0.37 | 0.25 | 0.64 | 0.09 | 0.36 | 0.70 | 0.64 | 0.51 |
| SC | Zunyi | 0.91 | 0.29 | 0.41 | 0.76 | 0.24 | 0.54 | 0.80 | 0.79 | 0.51 |
| SC | Yingtan | 0.63 | / | 0.07 | 0.35 | / | 0.31 | 0.22 | 0.42 | 0.18 |
| SC | Taoyuan | 0.83 | / | 0.53 | 0.82 | / | 0.56 | 0.76 | 0.78 | 0.40 |
| SC | Qianyanzhou | 0.87 | / | 0.12 | 0.59 | / | 0.40 | 0.63 | 0.59 | 0.54 |
| SC | Huitong | 0.81 | / | 0.45 | 0.77 | / | 0.69 | 0.80 | 0.78 | 0.45 |
| SC | Yanting | 0.71 | / | 0.32 | 0.43 | / | / | / | 0.72 | / |
| SC | Heshang | 0.77 | / | / | / | / | / | / | 0.73 | / |
| SC | Changsha | 0.82 | / | / | / | / | / | / | 0.79 | / |
| SE | Fuzhou | 0.89 | 0.24 | 0.09 | 0.50 | 0.24 | 0.23 | 0.37 | 0.52 | 0.33 |
| SE | Guangzhou | 0.90 | 0.36 | 0.60 | 0.60 | 0.47 | 0.61 | 0.58 | 0.44 | 0.44 |
| SE | Hong Kong | 0.80 | 0.61 | 0.60 | 0.79 | 0.64 | 0.68 | 0.78 | 0.64 | 0.62 |
| SE | Nanjing | 0.96 | 0.25 | 0.37 | 0.62 | 0.31 | 0.34 | 0.73 | 0.35 | 0.24 |
| SE | Changshu | 0.91 | / | 0.29 | 0.79 | / | 0.41 | 0.74 | 0.73 | 0.13 |
| SE | Dinghushan | 0.45 | / | 0.39 | 0.69 | / | 0.97 | 1.00 | 0.49 | 1.00 |
| SE | Ailaoshan | 0.97 | / | 0.66 | 0.87 | / | 0.70 | 0.83 | 0.14 | 0.80 |
| SE | Guangzhou | 0.74 | / | 0.47 | 0.84 | / | 0.91 | 0.83 | 0.64 | 0.77 |
| SE | Yongan | 0.43 | / | / | / | / | / | / | 0.35 | / |
| SE | Xishuangbanna | 0.96 | 0.53 | 0.82 | 0.85 | / | 0.57 | 0.78 | 0.68 | 0.78 |
| SE | Guangzhou | 0.77 | / | 0.51 | 0.83 | / | 0.79 | 0.76 | 0.73 | 0.64 |
| TP | Lhasa | 0.94 | 0.19 | 0.39 | 0.59 | 0.37 | 0.13 | 0.13 | 0.19 | 0.20 |

| Subregion | Station | Isoscape | CAM | GISSf | GISSn | HadAM | LMDZf | LMDZn | LMDZz | MIROC |
|---|---|---|---|---|---|---|---|---|---|---|
| TP | Lhasa | 0.78 | / | 0.58 | 0.59 | / | 0.36 | 0.19 | 0.27 | 0.45 |
| TP | Haibei | 0.87 | / | 0.24 | 0.30 | / | 0.64 | 0.50 | 0.43 | 0.40 |
| TP | Maoxian | 0.58 | / | -0.39 | 0.48 | / | 0.07 | 0.66 | 0.45 | 0.13 |
| TP | Gonggashan | 0.71 | / | 0.10 | 0.68 | / | 0.32 | 0.62 | 0.69 | 0.36 |
| TP | Delingha | 0.98 | 0.27 | 0.45 | 0.75 | 0.75 | 0.80 | 0.85 | 0.80 | 0.73 |
| TP | Nagqu | 0.92 | 0.17 | 0.16 | 0.64 | 0.15 | 0.15 | 0.06 | 0.46 | 0.40 |
| TP | Yushu | 0.81 | 0.00 | -0.12 | 0.37 | -0.11 | 0.18 | 0.28 | 0.46 | 0.29 |
| TP | Gaize | 0.96 | 0.34 | 0.03 | 0.44 | -0.06 | 0.36 | 0.37 | 0.43 | 0.18 |
| TP | Shiquanhe | 0.79 | -0.33 | 0.25 | 0.35 | -0.06 | 0.46 | 0.35 | 0.43 | 0.04 |
| TP | Lhasa | 0.94 | 0.37 | 0.63 | 0.74 | 0.45 | 0.45 | 0.40 | 0.42 | 0.56 |
| TP | Dingri | 0.89 | -0.08 | 0.61 | 0.18 | 0.71 | -0.33 | 0.44 | 0.37 | 0.35 |
| TP | Nyalam | 0.92 | 0.40 | 0.46 | 0.69 | -0.06 | 0.48 | 0.32 | 0.35 | 0.48 |
| TP | Tuotuohe | 0.94 | 0.18 | 0.10 | 0.64 | 0.44 | 0.66 | 0.67 | 0.67 | 0.48 |
| TP | Baidi | 0.84 | / | 0.51 | 0.62 | / | 0.29 | 0.47 | 0.44 | 0.49 |
| TP | Dui | 0.87 | / | 0.56 | 0.53 | / | 0.34 | 0.28 | 0.39 | 0.43 |
| TP | Taxkorgen | 0.97 | 0.19 | 0.70 | 0.79 | / | 0.80 | 0.82 | 0.83 | 0.70 |
| TP | Wengguo | 0.87 | / | 0.66 | 0.57 | / | 0.29 | 0.53 | 0.63 | 0.56 |
| TP | Lulang | 0.80 | / | 0.45 | 0.76 | / | 0.63 | 0.62 | 0.68 | 0.27 |
| TP | Nuxia | 0.59 | / | -0.21 | 0.79 | / | / | / | 0.49 | / |
| TP | Yeniugou | 0.84 | / | 0.88 | 0.83 | / | / | / | 0.88 | / |
| NW | Baotou | 0.93 | 0.43 | 0.52 | 0.51 | 0.39 | 0.43 | 0.60 | 0.28 | 0.46 |
| NW | Hetian | 0.96 | 0.49 | 0.52 | 0.82 | 0.55 | 0.81 | 0.89 | 0.88 | 0.82 |
| NW | Lanzhou | 0.94 | 0.18 | 0.22 | -0.07 | 0.53 | 0.60 | 0.60 | 0.45 | 0.61 |
| NW | Wulumuqi | 0.97 | 0.40 | 0.67 | 0.76 | 0.76 | 0.78 | 0.86 | 0.87 | 0.73 |
| NW | Yinchuan | 0.95 | 0.43 | 0.58 | 0.34 | 0.45 | 0.62 | 0.76 | 0.75 | 0.63 |
| NW | Zhangye | 0.96 | 0.35 | 0.59 | 0.57 | 0.62 | 0.80 | 0.83 | 0.79 | 0.76 |
| NW | Fukang | 0.94 | / | 0.75 | 0.78 | / | 0.87 | 0.88 | 0.86 | 0.84 |
| NW | Cele | 0.90 | / | -0.38 | 0.66 | / | 0.59 | 0.79 | 0.72 | 0.55 |
| NW | Linze | 0.93 | / | 0.68 | 0.67 | / | 0.78 | 0.79 | 0.68 | 0.81 |
| NW | Shapotou | 0.48 | / | 0.14 | 0.20 | / | 0.30 | 0.30 | -0.10 | 0.17 |
| NW | Ansai | 0.73 | / | -0.06 | 0.53 | / | -0.04 | 0.66 | 0.00 | 0.33 |
| NW | Erdos | 0.19 | / | 0.01 | 0.70 | / | 0.33 | 0.13 | 0.17 | -0.54 |
| NW | Naiman | 0.88 | / | 0.62 | 0.64 | / | 0.55 | 0.42 | 0.75 | 0.11 |

Table 3. Root mean square error (RMSE, ‰) metrics of $\delta^{18}O_p$ series between observations and iGCM simulations and the built isoscape at all stations.

| Subregion | Station | Isoscape | CAM | GISSf | GISSn | HadAM | LMDZf | LMDZn | LMDZz | MIROC |
|---|---|---|---|---|---|---|---|---|---|---|
| NE | Changchun | 7.47 | 11.50 | 14.71 | 14.98 | 17.35 | 16.14 | 11.45 | 22.16 | 13.80 |
| NE | Haerbin | 8.31 | 20.78 | 24.55 | 31.03 | 22.20 | 20.36 | 16.32 | 23.95 | 20.69 |
| NE | Qiqihar | 13.67 | 38.19 | 43.10 | 34.59 | 50.34 | 39.30 | 33.69 | 38.38 | 38.57 |
| NE | Changbaishan | 31.87 | / | 47.09 | 46.38 | / | 22.75 | 18.46 | 45.71 | 21.93 |
| NE | Sanjiang | 9.31 | / | 34.13 | 33.77 | / | 31.78 | 24.98 | 24.61 | 29.68 |
| NE | Hailun | 16.44 | / | 26.52 | 32.42 | / | 27.26 | 26.73 | 26.57 | 26.28 |
| NE | Shenyang | 20.70 | / | 24.61 | 20.10 | / | 22.11 | 12.67 | 33.25 | 17.39 |
| NC | Jinzhou | 5.46 | 10.88 | 8.25 | 6.59 | 7.97 | 6.56 | 6.68 | 21.52 | 11.30 |
| NC | Shijiazhuang | 15.12 | 56.29 | 49.40 | 45.68 | 40.41 | 44.44 | 36.17 | 83.95 | 52.45 |
| NC | Taiyuan | 5.37 | 22.94 | 23.43 | 24.56 | 20.93 | 19.33 | 20.78 | 33.76 | 19.09 |
| NC | Tianjin | 9.34 | 31.11 | 25.51 | 24.90 | 25.79 | 24.63 | 18.78 | 54.64 | 31.35 |
| NC | Xian | 10.27 | 40.99 | 35.42 | 32.86 | 32.12 | 28.35 | 20.92 | 30.46 | 26.61 |
| NC | Yantai | 5.84 | 19.21 | 21.43 | 18.19 | 18.76 | 20.83 | 17.23 | 36.96 | 24.88 |
| NC | Zhengzhou | 10.99 | 36.21 | 32.11 | 23.59 | 26.80 | 25.12 | 20.52 | 42.04 | 23.46 |
| NC | Beijing | 28.37 | / | 20.28 | 27.08 | / | 14.61 | 14.19 | 29.14 | 10.71 |
| NC | Fengqiu | 10.03 | / | 22.74 | 17.20 | / | 18.24 | 14.66 | 27.88 | 19.44 |
| NC | Yucheng | 13.78 | / | 21.10 | 19.83 | / | 17.29 | 15.57 | 31.43 | 15.38 |
| NC | Changwu | 13.79 | / | 40.64 | 37.52 | / | 21.68 | 13.69 | 31.02 | 28.30 |
| SC | Changsha | 9.74 | 21.83 | 23.99 | 22.08 | 27.34 | 19.65 | 15.59 | 19.36 | 19.72 |
| SC | Chengdu | 16.55 | 56.26 | 56.99 | 62.06 | 39.34 | 29.57 | 25.76 | 26.21 | 43.54 |
| SC | Guilin | 10.18 | 24.79 | 24.69 | 19.40 | 29.09 | 21.52 | 19.23 | 18.46 | 21.60 |
| SC | Guiyang | 9.14 | 31.95 | 26.87 | 23.28 | 30.90 | 25.78 | 19.23 | 22.24 | 25.05 |
| SC | Kunming | 16.06 | 47.73 | 44.50 | 44.59 | 44.75 | 47.67 | 40.91 | 47.68 | 45.03 |
| SC | Liuzhou | 11.06 | 20.81 | 19.86 | 16.67 | 21.12 | 18.23 | 16.09 | 17.71 | 16.44 |
| SC | Wuhan | 7.79 | 19.69 | 24.96 | 19.37 | 24.38 | 20.34 | 15.64 | 16.61 | 17.62 |
| SC | Zunyi | 13.82 | 38.47 | 33.45 | 31.05 | 34.47 | 28.45 | 21.06 | 22.14 | 28.87 |
| SC | Yingtan | 17.72 | / | 22.76 | 25.17 | / | 13.90 | 19.02 | 22.05 | 14.57 |
| SC | Taoyuan | 11.19 | / | 17.54 | 15.16 | / | 13.78 | 11.46 | 14.23 | 16.38 |
| SC | Qianyanzhou | 7.51 | / | 20.86 | 18.09 | / | 14.37 | 13.41 | 13.93 | 13.33 |
| SC | Huitong | 12.06 | / | 18.17 | 15.19 | / | 10.52 | 9.30 | 14.60 | 15.35 |
| SC | Yanting | 21.13 | / | 31.54 | 33.49 | / | / | / | 20.09 | / |
| SC | Heshang | 20.75 | / | / | / | / | / | / | 21.97 | / |
| SC | Changsha | 19.15 | / | / | / | / | / | / | 21.62 | / |
| SE | Fuzhou | 10.88 | 27.18 | 28.32 | 22.42 | 27.09 | 28.15 | 25.79 | 21.75 | 25.71 |
| SE | Guangzhou | 6.39 | 15.12 | 10.76 | 10.91 | 16.41 | 11.04 | 13.11 | 17.21 | 13.25 |
| SE | Hong Kong | 38.15 | 43.96 | 47.98 | 34.52 | 50.52 | 42.01 | 32.63 | 45.53 | 40.70 |
| SE | Nanjing | 6.91 | 22.14 | 19.21 | 17.01 | 23.01 | 25.58 | 20.30 | 22.12 | 24.79 |
| SE | Changshu | 3.50 | / | 9.88 | 5.73 | / | 14.49 | 10.67 | 6.84 | 11.59 |
| SE | Dinghushan | 22.85 | / | 15.51 | 14.40 | / | 3.71 | 1.20 | 22.44 | 5.39 |
| SE | Ailaoshan | 6.46 | / | 20.21 | 15.92 | / | 27.27 | 24.72 | 33.50 | 25.93 |
| SE | Guangzhou | 20.37 | / | 16.56 | 9.74 | / | 5.71 | 6.20 | 23.52 | 6.89 |
| SE | Yongan | 15.83 | / | / | / | / | / | / | 16.42 | / |
| SE | Xishuangbanna | 9.26 | 20.69 | 19.55 | 16.55 | / | 32.37 | 29.43 | 28.11 | 32.21 |
| SE | Guangzhou | 11.43 | / | 16.95 | 11.17 | / | 9.01 | 8.77 | 12.06 | 9.43 |
| TP | Lhasa | 16.48 | 45.39 | 48.24 | 40.56 | 42.23 | 50.30 | 53.37 | 43.84 | 46.20 |

| Subregion | Station | Isoscape | CAM | GISSf | GISSn | HadAM | LMDZf | LMDZn | LMDZz | MIROC |
|---|---|---|---|---|---|---|---|---|---|---|
| TP | Lhasa | 24.27 | / | 30.61 | 30.51 | / | 34.27 | 37.35 | 36.78 | 30.10 |
| TP | Haibei | 11.23 | / | 38.38 | 45.39 | / | 21.11 | 19.96 | 33.95 | 29.98 |
| TP | Maoxian | 13.61 | / | 25.84 | 29.51 | / | 13.38 | 9.50 | 15.67 | 12.89 |
| TP | Gonggashan | 27.37 | / | 26.75 | 31.77 | / | 26.49 | 24.21 | 30.24 | 23.97 |
| TP | Delingha | 14.94 | 87.68 | 127.83 | 113.21 | 53.20 | 40.92 | 35.78 | 85.71 | 71.41 |
| TP | Nagqu | 16.24 | 40.61 | 42.90 | 35.18 | 25.44 | 61.90 | 62.31 | 40.53 | 35.25 |
| TP | Yushu | 11.73 | 32.84 | 50.28 | 37.89 | 13.08 | 31.88 | 29.51 | 20.69 | 29.43 |
| TP | Gaize | 13.95 | 44.70 | 72.06 | 53.95 | 31.61 | 51.14 | 50.41 | 40.32 | 47.10 |
| TP | Shiquanhe | 20.98 | 37.84 | 33.23 | 32.83 | 30.81 | 50.83 | 49.15 | 35.91 | 51.55 |
| TP | Lhasa | 22.86 | 48.17 | 51.15 | 47.12 | 41.72 | 75.87 | 79.16 | 57.43 | 61.10 |
| TP | Dingri | 9.82 | 20.02 | 17.23 | 25.29 | 13.54 | 41.22 | 40.39 | 20.09 | 39.40 |
| TP | Nyalam | 22.22 | 44.29 | 48.22 | 37.36 | 65.87 | 56.07 | 58.01 | 52.53 | 64.70 |
| TP | Tuotuohe | 22.46 | 84.32 | 112.41 | 78.12 | 48.50 | 56.33 | 55.78 | 45.39 | 60.21 |
| TP | Baidi | 12.24 | / | 21.66 | 20.40 | / | 33.47 | 32.83 | 23.63 | 36.25 |
| TP | Dui | 17.57 | / | 28.75 | 28.00 | / | 43.04 | 43.72 | 34.06 | 44.76 |
| TP | Taxkorgen | 12.64 | 13.27 | 44.91 | 48.34 | / | 31.72 | 30.37 | 32.65 | 34.84 |
| TP | Wengguo | 10.97 | / | 15.37 | 17.58 | / | 29.67 | 29.66 | 19.67 | 31.43 |
| TP | Lulang | 39.66 | / | 29.51 | 25.03 | / | 27.43 | 26.70 | 44.52 | 22.53 |
| TP | Nuxia | 25.64 | / | 10.31 | 14.28 | / | / | / | 24.23 | / |
| TP | Yeniugou | 16.60 | / | 30.61 | 37.42 | / | / | / | 16.30 | / |
| NW | Baotou | 11.33 | 50.59 | 42.77 | 38.48 | 43.78 | 32.82 | 24.41 | 60.87 | 40.64 |
| NW | Hetian | 19.61 | 60.60 | 67.10 | 77.90 | 76.83 | 36.28 | 31.10 | 61.56 | 38.01 |
| NW | Lanzhou | 10.60 | 55.48 | 53.01 | 65.13 | 28.74 | 24.34 | 23.92 | 38.27 | 42.90 |
| NW | Wulumuqi | 17.15 | 66.80 | 59.34 | 69.90 | 63.51 | 55.43 | 44.56 | 69.41 | 49.72 |
| NW | Yinchuan | 8.14 | 38.30 | 33.31 | 35.84 | 27.23 | 21.02 | 18.36 | 33.55 | 29.89 |
| NW | Zhangye | 20.80 | 84.71 | 91.43 | 93.86 | 53.70 | 40.24 | 37.27 | 65.15 | 64.30 |
| NW | Fukang | 17.91 | / | 35.66 | 46.33 | / | 23.33 | 21.36 | 40.93 | 23.37 |
| NW | Cele | 8.46 | / | 49.74 | 52.79 | / | 18.44 | 17.65 | 32.59 | 25.31 |
| NW | Linze | 16.52 | / | 48.37 | 51.71 | / | 21.65 | 21.60 | 47.16 | 30.30 |
| NW | Shapotou | 18.98 | / | 41.20 | 52.35 | / | 14.44 | 16.28 | 41.61 | 23.84 |
| NW | Ansai | 18.09 | / | 38.55 | 32.02 | / | 24.17 | 13.92 | 37.28 | 23.82 |
| NW | Erdos | 14.76 | / | 22.65 | 20.64 | / | 7.69 | 9.35 | 24.33 | 11.91 |
| NW | Naiman | 12.76 | / | 13.85 | 13.75 | / | 12.61 | 12.42 | 36.06 | 14.49 |

[Figure]

Figure 3. Spatial distributions of $\delta^{18}O_p$ in each month for the period of 1979-2007 as obtained from observations (circles), the built isoscape (left column), and better-performing iGCMs (right three columns).

[Figure]

Figure 3. Spatial distributions of δ¹⁸Oₚ in each month for the period of 1979-2007 as obtained from observations (circles), the built isoscape (left column), and better-performing iGCMs (right three columns).

---

## Author Comment (AC4)

**Responses to RC2:**

*As one of the key tracers of hydroclimate change, precipitation isotopes have very important research significance. According to the current situation of less observational data, the authors used iGCMs to obtain high-resolution precipitation isotope data over the past 148 years, which will provide important data support for the study of precipitation isotopes and hydroclimate change. I think this research is meaningful, but there are still some problems in the article, I think it needs to be further elaborated before accepting publication.*

**Re:** We would like to thank the reviewer for the comments and suggestions, which helped us improve the quality of our work. We have carefully revised the manuscript, especially for the validation of the dataset's quality, adding more details and enriching the results. Please find a detailed point-by-point response to each comment.

*1. There is a problem with regional division. For example, Yunnan Province is a typical southwestern region of China, and its climate is dominantly influenced by the Indian summer monsoon, which is different from the region of southeastern China where is mainly influenced the East Asian summer monsoon. How can it be divided into southeastern regions? Generally, the Indian summer monsoon precipitation oxygen isotope values during JJAS are lower than the East Asian summer monsoon rainfall. I don't think current regional division is scientific.*

**Re:** Thanks for the valuable comments. We agree with the reviewer that some current sub-region division is not fully appropriate, because we have tried to include as many stations as possible within a sub-region to train the data fusion and bias correction methods. According to the reviewer's suggestions, we have re-divided the sub-regions according to review's suggestions (Fig. 1) by taking terrain and climatic conditions into account, as well as ensuring sufficient number of stations and data volumes in a sub-region.

[Figure]

Figure 1. New sub-region division (NE – Northeast China, NC – North China, SE – Southeast China, SW – Southwest China, TP – Tibetan Plateau, NW – Northwest China).

*2. There are some GNIP stations with long-term precipitation isotope monitoring data, which can be used to compare long-term changes with the simulation results, such as the Hong Kong station. I suggest that the authors can compare the Hong Kong station and other stations with 8-10 years of monitoring data with the simulated isotope records. This comparison can be used to verify the reliability of the simulation.*

**Re:** Thanks for the suggestion. Two stations with appropriate length of observation were randomly selected for each sub-region. Totally, 12 stations were selected. The time series of $\delta^{18}O_p$ were plotted for observations, iGCM simulations, and the generated isoscape (Fig. 2). As can be seen from Fig. 2, the variations of $\delta^{18}O_p$ are very consistent between the generated isoscape and observations, and the isoscape performs much better than raw iGCM simulations. In particular for the period before 2007, the CNN model integrates the advantages of various simulations and captures most features of the observed data. These results generally prove that the generated isoscape is reliable.

*3. At present, the altitudes of each monitoring site vary greatly. When the author uses the monitoring data of each site for spatial analysis and compares them with the simulation results, did the author consider the effect of altitude on precipitation isotopes to calibrate? For example, according to the relationship between altitude and precipitation isotopes at each site or the large region scale, first calibrate the precipitation isotope data of all stations to the same altitude, and then compare it with the simulated data. Because the simulation precipitation isotope results are at the same altitude, if no calibration is performed, the spatial comparison of the simulated and monitored precipitation isotope data will inevitably not be the real results. In the current manuscript, it seems that the author has not calibrated, and it is suggested that the author add relevant correction processes or solutions.*

**Re:** We agree with the reviewer that altitude is an important factor to be considered in the study. However, we do not think that iGCMs as GCMs are simulated at the same altitude. The sigma vertical coordinate is commonly used in climate models (Schmidt et al., 2006; Hourdin et al., 2006; Collins et al., 2004), in which the lowest coordinate surface follows the model terrain, and pressure at other layers are scaled with the surface pressure. The precipitation isotope data from the surface layer were used in our study. Moreover, we are afraid the use of the direct relationship between altitude and precipitation isotopes for calibration may introduce uncertainty or bias to dataset, because the precipitation isotopes in a region are affected by latitude, land and sea location, temperature, precipitation and many other factors. However, in the revised manuscript, we will introduce the physical-based ancillary data such as elevation and meteorological data in the fusion processes. The details of the introduced data are Shuttle Radar Topography Mission digital elevation data (SRTM DEM) with the spatial resolution of 90m, and dataset of gridded monthly precipitation and temperature in China with spatial resolution of the 0.5 degree.

References:

Collins, W.D., Rasch, P.J., Boville, B.A., Hack, J.J., McCaa, J.R., Williamson, D.L., Kiehl, J.T., Briegleb, B., Bitz, C., Lin, S.J. and Zhang, M.: Description of the NCAR community atmosphere model (CAM 3.0), NCAR Tech. Note NCAR/TN-464+ STR, 226, 1326-1334, 2004.

Hourdin, F., Musat, I., Bony, S., Braconnot, P., Codron, F., Dufresne, J.L., Fairhead, L., Filiberti, M.A., Friedlingstein, P., Grandpeix, J.Y. and Krinner, G.: The LMDZ4 general circulation model: climate performance and sensitivity to parametrized physics with emphasis on tropical convection, Climate Dynamics, 27, 787-813,

https://doi.org/10.1007/s00382-006-0158-0, 2006.

Schmidt, G.A., Ruedy, R., Hansen, J.E., Aleinov, I., Bell, N., Bauer, M., Bauer, S., Cairns, B., Canuto, V., Cheng, Y. and Del Genio, A.: Present-day atmospheric simulations using GISS ModelE: Comparison to in situ, satellite, and reanalysis data, Journal of Climate, 19, 153-192, https://doi.org/10.1175/JCLI3612.1, 2006.

[Figure]

Figure 2. Time-series comparisons of $\delta^{18}O_p$ among the built isoscape, iGCM simulations, and in-situ observations at selected stations in each sub-region.

[Figure]

Figure 2. Time-series comparisons of δ18Op among the built isoscape, iGCM simulations, and in-situ observations at selected stations in each sub-region.

[Figure]

Figure 2. Time-series comparisons of $\delta^{18}O_p$ among the built isoscape, iGCM simulations, and in-situ observations at selected stations in each sub-region.